# Catecholaminergic neuromodulation and selective attention jointly shape perceptual decision-making

**Stijn A Nuiten[1,2,3]\*, Jan Willem de Gee[2,4,5,6], Jasper B Zantvoord[7,8], Johannes J Fahrenfort[1,2,9,10], Simon van Gaal[1,2]\***

[1]Department of Psychology, University of Amsterdam, Amsterdam, Netherlands; [2]Amsterdam Brain & Cognition, University of Amsterdam, Amsterdam, Netherlands; [3]Department of Psychiatry (UPK), University of Basel, Basel, Switzerland; [4]Jan and Dan Duncan Neurological Research Institute, Texas Children's Hospital, Houston, United States; [5]Department of Neuroscience, Baylor College of Medicine, Houston, United States; [6]Cognitive and Systems Neuroscience, Swammerdam Institute for Life Sciences, University of Amsterdam, Amsterdam, Netherlands; [7]Department of Psychiatry, Amsterdam UMC location University of Amsterdam, Amsterdam, Netherlands; [8]Amsterdam Neuroscience, Amsterdam, Netherlands; [9]Institute for Brain and Behavior Amsterdam, Vrije Universiteit Amsterdam, Amsterdam, Netherlands; [10]Department of Experimental and Applied Psychology - Cognitive Psychology, Vrije Universiteit Amsterdam, Amsterdam, Netherlands

\*For correspondence:
stijnnuiten@gmail.com (SAN);
s.vangaal@uva.nl (SvG)

**Competing interest:** The authors declare that no competing interests exist.

**Abstract** Perceptual decisions about sensory input are influenced by fluctuations in ongoing neural activity, most prominently driven by attention and neuromodulator systems. It is currently unknown if neuromodulator activity and attention differentially modulate perceptual decision-making and/or whether neuromodulatory systems in fact control attentional processes. To investigate the effects of two distinct neuromodulatory systems and spatial attention on perceptual decisions, we pharmacologically elevated cholinergic (through donepezil) and catecholaminergic (through atomoxetine) levels in humans performing a visuo-spatial attention task, while we measured electroencephalography (EEG). Both attention and catecholaminergic enhancement improved decision-making at the behavioral and algorithmic level, as reflected in increased perceptual sensitivity and the modulation of the drift rate parameter derived from drift diffusion modeling. Univariate analyses of EEG data time-locked to the attentional cue, the target stimulus, and the motor response further revealed that attention and catecholaminergic enhancement both modulated pre-stimulus cortical excitability, cue- and stimulus-evoked sensory activity, as well as parietal evidence accumulation signals. Interestingly, we observed both similar, unique, and interactive effects of attention and catecholaminergic neuromodulation on these behavioral, algorithmic, and neural markers of the decision-making process. Thereby, this study reveals an intricate relationship between attentional and catecholaminergic systems and advances our understanding about how these systems jointly shape various stages of perceptual decision-making.

## eLife assessment

This **important** study shows that pharmacologically enhanced catecholamine levels and increased voluntary spatial attention have overlapping as well as dissociable effects on performance on a visuospatial attention task and corresponding EEG markers. The findings provide **solid** evidence

regarding how neuromodulatory arousal and selective spatial attention jointly shape perceptional decision-making.

## Introduction

Agents often make inconsistent and varying decisions when faced with identical repetitions of (sensory) evidence. Recent laboratory experiments, meticulously controlling for external factors, have shown that ongoing fluctuations in neural activity may underlie such behavioral variability (*Gold and Shadlen, 2007*; *Renart and Machens, 2014*; *Waschke et al., 2021*; *Wyart and Koechlin, 2016*). Variability in behavioral responses, however, may not be exclusively related to alterations in the decision-making process, but in fact may find its root cause in alterations in sensory processes leading up toward the decision (*Benwell et al., 2017*; *Busch et al., 2009*; *Iemi et al., 2017*; *Iemi et al., 2022*; *Samaha et al., 2020*).

Candidate sources of these fluctuations in cortical activity and behavior are systematic variations in attention and central arousal state (*Harris and Thiele, 2011*; *Summerfield and Egner, 2009*). Attention to specific features of sensory input (e.g. its spatial location or content) results in facilitated processing of this input and is associated with improved behavioral performance (*Carrasco, 2011*; *Desimone and Duncan, 1995*). The central arousal state of animals is controlled by the activity of neuromodulators, including noradrenaline (NA; *Aston-Jones and Cohen, 2005*) and acetylcholine (ACh; *Hasselmo and Sarter, 2011*; *McCormick, 1989*), which globally innervate cortex. Contrary to attention, the relation between noradrenergic and cholinergic neuromodulator activity and behavioral performance is non-monotonic, with optimal behavioral performance occurring at intermediate levels of neuromodulation (*Aston-Jones and Cohen, 2005*; *Bentley et al., 2011*; *McGinley et al., 2015a*). It is currently unknown if fluctuations in attention and neuromodulator activity contribute to neural and behavioral variability in the same way. Both attention and neuromodulators can alter the input/output ratio (or: gain) of single neurons and neuronal networks, thereby modulating the impact of (sensory) input on cortical processing (*Aston-Jones and Cohen, 2005*; *Humphreys et al., 1998*; *Soma et al., 2012*) and thus possibly perceptual experience. For example, recordings in macaque middle temporal area have provided evidence for attentional modulation of neural gain, by showing increased firing rates for neurons having their receptive field on an attended location in space, but suppressed firing rates for neurons with receptive fields outside that focus, under identical sensory input (*Martinez-Trujillo and Treue, 2004*). Likewise, both NA and ACh are known to modulate firing rates of neurons in sensory cortices (*McCormick, 1989*). To illustrate, the arousal state of mice, driven by neuromodulator activity, strongly modulates ongoing membrane potentials in auditory cortex and as such can shape optimal states for auditory detection behavior (*McGinley et al., 2015a*). Neuromodulator activity and attention also have similar effects on network dynamics, as both increase cortical desynchronization and enhance the encoding of sensory information (*Harris and Thiele, 2011*; *Thiele and Bellgrove, 2018*). These similar neural effects of attention and neuromodulators have raised the question whether neuromodulatory systems control (certain aspects of) attention (*Thiele and Bellgrove, 2018*).

Here, we addressed this question by investigating if and how covert spatial attention and neuromodulatory systems jointly shape, in isolation and/or in interaction, visual perception in humans. To explore how fluctuations in neuromodulatory drive affect perceptual and decision-making processes in humans, previous work has so far mainly used correlational methods, often by linking performance on simple discrimination or detection tasks to different readouts of fluctuations in brain state, most prominently variations in pupil size (*de Gee et al., 2017*; *de Gee et al., 2014*; *Murphy et al., 2014*; *Podvalny et al., 2021*; *van Kempen et al., 2019*; *Waschke et al., 2019*; for exceptions see, e.g., *Beste et al., 2018*; *Gelbard-Sagiv et al., 2018*; *Loughnane et al., 2019*). To provide causal evidence directly tying neuromodulatory drive to attention, we pharmacologically elevated levels of catecholamines (NA and dopamine, through atomoxetine [ATX]) and ACh (through donepezil [DNP]) in human participants performing a probabilistic attentional cueing task, while we measured electroencephalography (EEG) and pupillary responses. Participants reported the orientation of a briefly presented Gabor patch, presented left or right of fixation, as being clockwise (45°) or counterclockwise (–45°). A visual cue predicted the location of the Gabor with 80% validity (the cue did not predict orientation, see *Figure 1A*). This set-up allowed us to test the effects of increased neuromodulator levels (drug effects) and spatial attention (cue validity effects) on several stages of cortical processing leading

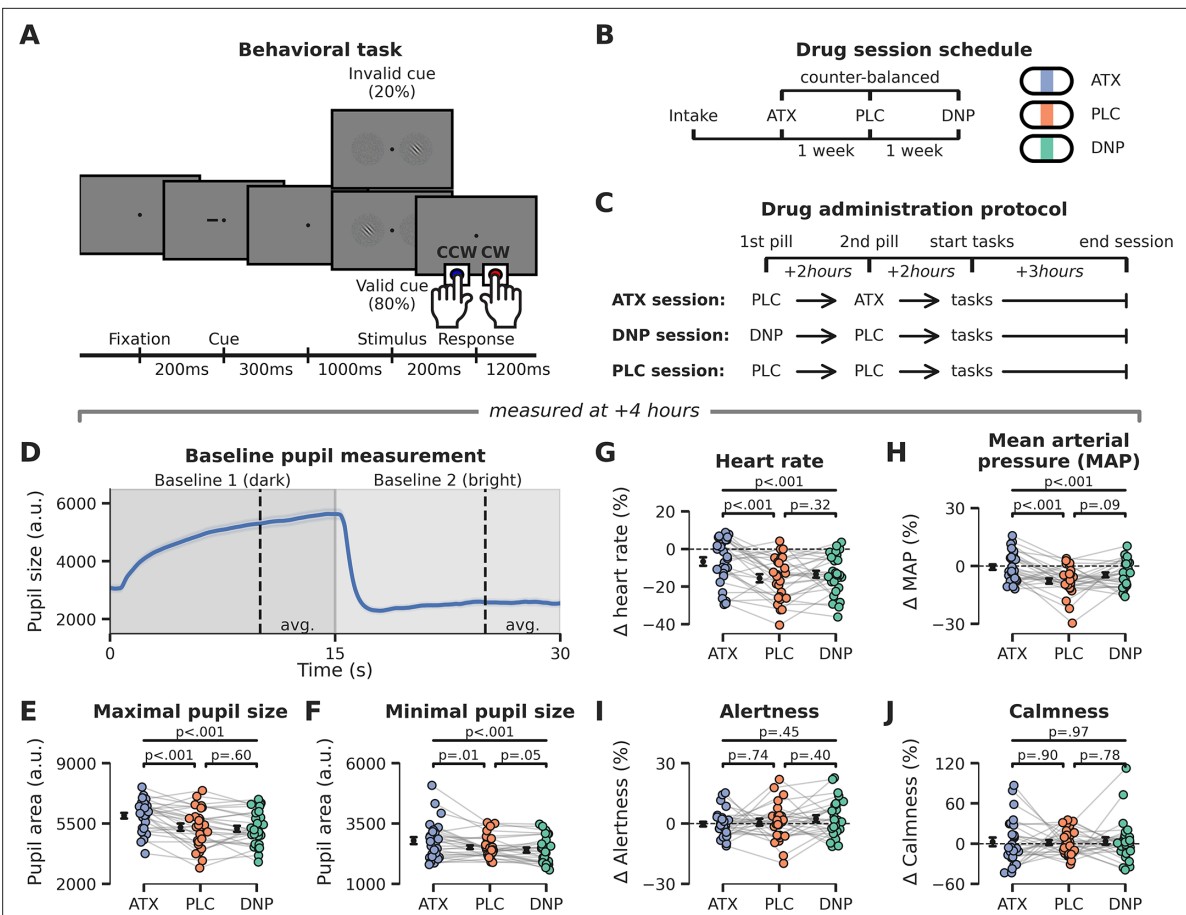

**Figure 1.** Experimental set-up: behavioral task, pharmacological manipulation, and physiological responses. (**A**) Schematic representation of the behavioral task. Participants responded to the orientation (clockwise/counterclockwise [CW/CCW]) of unilaterally presented Gabor stimuli that were embedded in noise (bilaterally presented). The likely location of the Gabor stimulus was cued (horizontal dash presented 0.33° left/right from fixation) with 80% validity before stimulus onset. (**B**) Schematic overview of experimental sessions. Participants came to the lab on four occasions: one intake session and three experimental sessions. On the experimental sessions, participants received either placebo (PLC, data in orange), donepezil (DNP, 5 mg, data in green), or atomoxetine (ATX, 40 mg, data in blue). Drug order was counterbalanced across participants. (**C**) Time schedule of experimental sessions. Participants received a pill on two moments in each session, one at the beginning of the session and a second pill 2 hr later. The first pill contained either placebo (PLC and ATX session) or donepezil (DNP session), the second pill was either a placebo (PLC and DNP session) or atomoxetine (ATX session). Behavioral testing started 4 hr after administration of the first pill. (**D**) Baseline pupil diameter was measured before onset of the behavioral task. Participants fixated while the background luminance of the monitor was dimmed (for 15 s) and then brightened (for 15 s) to establish the pupil size in dark and bright circumstances. Shading indicates standard error of the mean (SEM). (**E–J**) Effects of drug on pupil diameter during the dark ($P_{max}$, **E**) and bright ($P_{min}$, **F**) measurement windows, heart rate (HR, **G**), mean arterial blood pressure (MAP, **H**, see Materials and methods), and subjective ratings (visual analogue scale [VAS]; see Materials and methods) of alertness (panel **I**) and calmness (**J**). All measurements except pupil diameter were baseline-corrected to the first measurement taken right before ingestion of the first pill.

The online version of this article includes the following source data and figure supplement(s) for figure 1:

**Source data 1.** Source files for physiological data.

**Figure supplement 1.** Late effects of drug on subjective and bodily measures of arousal.

**Figure supplement 1—source data 1.** Source files for late physiological data.

up to the perceptual decision about Gabor orientation. We first characterized the effects of drug condition and cue validity on perceptual sensitivity, derived from signal detection theory (SDT) (*Green and Swets, 1966*) and latent decision parameters, derived from drift diffusion modeling (*Ratcliff and McKoon, 2008*). We then turned to neural data that was locked to three relevant events: cue presentation, stimulus presentation, and the behavioral response. Specifically, we investigated the effects of attention and neuromodulation on accumulation of sensory evidence over time toward a decision threshold (*Loughnane et al., 2019*; *Nieuwenhuis et al., 2005*; *van Kempen et al., 2019*),

stimulus-evoked sensory activity (*Loughnane et al., 2016*; *Newman et al., 2017*; *van Kempen et al., 2019*), and prestimulus anticipatory neural signals (*Bauer et al., 2012*; *Dahl et al., 2020*; *Praamstra and Kourtis, 2010*; *van Velzen and Eimer, 2003*).

## Results

### Catecholaminergic enhancement increases physiological markers of bodily arousal

We employed a randomized, double-blind crossover design in which ATX (40 mg), DNP (5 mg), and placebo (PLC) were administered in different EEG recording sessions (drug order was counterbalanced between participants, *Figure 1B*). ATX is a relatively selective NA reuptake inhibitor, which inhibits the presynaptic NA reuptake transporter, thereby resulting in increased NA and dopamine levels (*Simpson and Plosker, 2004*). DNP is a cholinesterase inhibitor, which impedes the breakdown of ACh by cholinesterase, thereby resulting in overall increased ACh levels (*Tiseo et al., 1998*). Our drug administration procedure was set up to take the difference in time to reach maximum plasma concentrations ($T_{max}$) of DNP (~4 hr) and ATX (~2 hr) into account, while keeping participants blind to the current drug condition. Because $T_{max}$ was different for ATX and DNP and we did not want to introduce any differences in terms of the pill-taking regime, we added a PLC pill within all the experimental sessions, also for the ATX and the DNP protocol. Thus, participants would always ingest pills at the same moments in time, regardless of the session they were in. On every experimental session, we always administered two pills at the same moments in time, differing only in which of these two pills contained the drug. Specifically, in the DNP session this meant that the first pill was DNP followed 2 hr later by PLC, while for the ATX session this meant that the first pill was PLC followed 2 hr later by ATX. In the PLC session both pills were PLC (see *Figure 1C* below).

To gauge the effect of our pharmaceuticals, we collected physiological and subjective state measures at different moments throughout the day (from 9:00 to 16:00; *Figure 1D–J*; for details see Materials and methods). We performed one-way repeated measures (rm)ANOVAs on these physiological and subjective measures, to test for omnibus main effects of drug condition. Next, we used post hoc paired-sample t-tests to test for pairwise drug effects (ATX/DNP vs. PLC). Throughout this work, all effect sizes are expressed as $\eta_p^2$ and should be interpreted as medium sized for $\eta_p^2 > 0.06$ and large for $\eta_p^2 > 0.14$ (*Cohen, 1988*).

Prior to onset of the behavioral tasks, we observed strong effects of drug condition on pupil size ($P_{max}$: $F_{2,54} = 15.84$, p<0.001, $\eta_p^2 = 0.37$; $P_{min}$: $F_{2,54} = 10.56$, p<0.001, $\eta_p^2 = 0.28$), heart rate ($F_{2,54} = 8.36$, p<0.001, $\eta_p^2 = 0.24$), and mean arterial blood pressure ($F_{2,54} = 8.05$, p<0.001, $\eta_p^2 = 0.23$). Post hoc tests indicated that all physiological measures were only affected by ATX ($P_{max}$: t(27)=4.12, p<0.001, $\eta_p^2 = 0.39$; $P_{min}$: t(27)=2.77, p=0.01, $\eta_p^2 = 0.22$; HR: t(27)=4.11, p<0.001, $\eta_p^2 = 0.38$; MAP: t(27)=4.33, p<0.001, $\eta_p^2 = 0.41$) and not DNP ($P_{max}$: t(27)=–0.52, p=0.60, $\eta_p^2 = 0.01$, $BF_{01}=4.11$; $P_{min}$: t(27)=–2.02, p=0.05, $\eta_p^2 = 0.13$, note this is a trend in the opposite direction; HR: t(27)=1.01, p=0.32, $\eta_p^2 = 0.04$, $BF_{01}=3.13$; MAP: t(27)=1.76, p=0.09, $\eta_p^2 = 0.10$, $BF_{01}=0.78$). The robust effects of ATX, but absence of effects under DNP are in line with previous non-clinical reports (*Pfeffer et al., 2018*; *Pfeffer et al., 2021*). Moreover, the effects of ATX on physiological measures of arousal were long lasting (*Figure 1—figure supplement 1*). We performed an additional multivariate ANOVA to test whether grouping the physiological responses (HR, MAP, and pupil) would elucidate global effects of DNP, but this was not the case (Wilks' lambda = 0.93, $F_{5,50}=0.99$, p=0.42). Drug condition did not modulate subjective measures of alertness ($F_{2,54}=0.82$, p=0.45, $\eta_p^2 = 0.03$) and calmness ($F_{2,54}=0.03$, p=0.97, $\eta_p^2 = 0.00$), suggesting that participants were not aware of their heightened arousal state. However, forced-choice guessing at the end of the day about receiving any active substance (ATX, DNP) or not (PLC) suggested that some participants may have been aware of being in a drug session (proportion z-tests ATX vs. PLC: z=2.96, p=0.003; DNP vs. PLC: z=0.55, p=0.10).

### Catecholaminergic enhancement and attention modulate the rate of sensory evidence accumulation

To establish if and how cue validity and drug condition affected perceptual decision-making, we tested their respective effects on both the outcomes and latent constituents of the decision-making process. First, we report the effects of cue validity and drug on outcome measures of the decision,

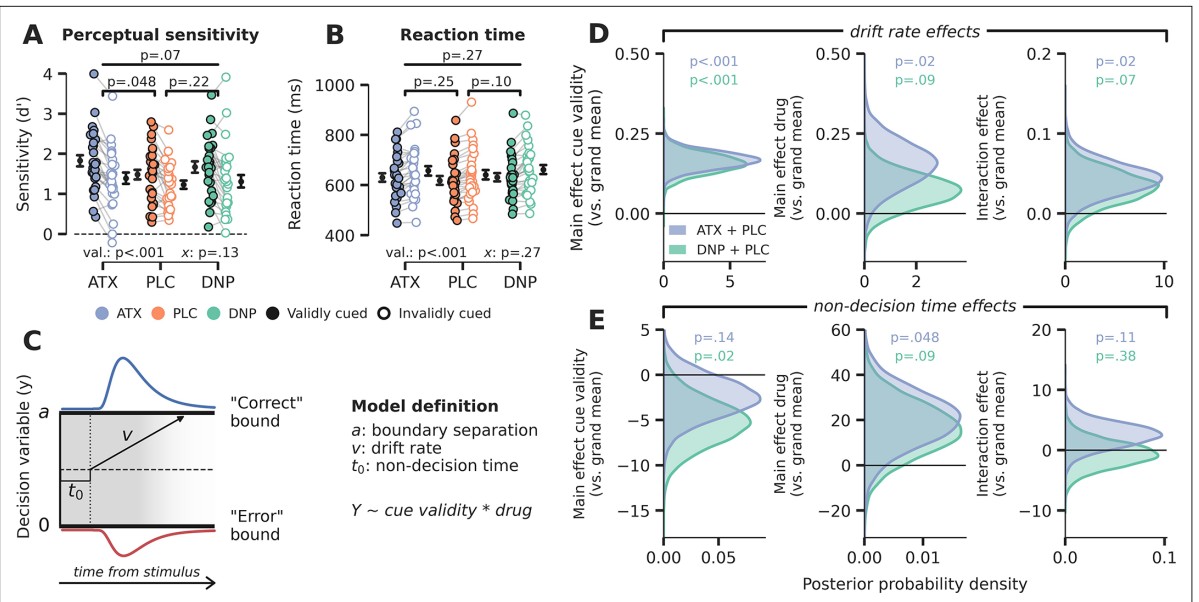

**Figure 2.** Behavioral results. (**A**) Signal detection theoretic sensitivity (**d'**), separately per drug and cue validity. Error bars indicate SEM, x demarks the p-value for the omnibus interaction effect between drug condition and cue validity, val. refers to the factor cue validity. (**B**) As A, but for reaction time (RT). (**C**) Schematic of the drift diffusion model (DDM), accounting for behavioral performance and RTs. The model describes behavior based on various latent parameters, including drift rate (**v**), boundary separation (**a**), and non-decision time (**t₀**). These parameters (demarked with Y in formula) were allowed to fluctuate with cue validity, drug condition, and their interaction. Models were fitted separately for atomoxetine (ATX) (+placebo [PLC]) and donepezil (DNP) (+PLC). (**D–E**) Posterior probability distributions for DDM parameter estimates (blue: ATX model, green: DNP model). The effects of cue validity (left column), drug (middle column), and their interaction (right column) are shown for (**D**) drift rate and (**E**) non-decision time.

The online version of this article includes the following source data and figure supplement(s) for figure 2:

**Source data 1.** Source files for behavioral data and computational analyses.

**Figure supplement 1.** Cue validity and drug condition effects on (choice history) bias.

**Figure supplement 1—source data 1.** Source files for behavioral data.

**Figure supplement 2.** HDDM model fits.

**Figure supplement 2—source data 1.** Source files for drift diffusion model (DDM) fits.

**Figure supplement 3.** No effects of cue validity, drug, and their interaction on decision bound separation in weighted regression drift diffusion model (DDM).

**Figure supplement 3—source data 1.** Source files for drift diffusion model (DDM) model bound estimates.

**Figure supplement 4.** Parameter estimates of unweighted regression drift diffusion model (DDM).

**Figure supplement 4—source data 1.** Source files for unweighted drift diffusion model (DDM) regression model parameter estimates.

**Figure supplement 5.** Drift rate variability was not modulated by drug or cue validity.

**Figure supplement 5—source data 1.** Source files for weighted drift diffusion model (DDM) regression drift rate variability model parameter estimate.

quantified with perceptual sensitivity (d'), derived from SDT (**Green and Swets, 1966**), and reaction times (RTs). Specifically, first we used 3×2 factor (drug × cue validity) rmANOVAs to test for omnibus effects of cue validity, drug condition, and their interactions (leveraging the full scope of data in our design). Thereafter, we performed planned 2×2 factor (specific drug × cue validity) rmANOVAs to test for pairwise main and interaction effects of each active drug (ATX/DNP vs. PLC, see Materials and methods for detailed description of statistical analyses). These drug-specific ANOVAs test for drug modulations that may be obscured in the full factor ANOVA.

As expected, validly cued (attended) targets were associated with improved d' and increased response speed compared to invalidly cued (unattended) targets (main effects, d': $F_{1,27}=39.22$, $p<0.001$, $\eta_p^2 = 0.59$; RT: $F_{1,27}=43.47$, $p<0.001$, $\eta_p^2 = 0.62$). This beneficial effect of cue validity on d' was not modulated by drug ($F_{2,54}=2.11$, $p=0.13$, $\eta_p^2 = 0.07$. $BF_{01}=1.87$; **Figure 2B**), and was present in both drug conditions, although post hoc tests showed a trending interaction between cue validity and ATX vs PLC ($F_{1,27}=3.76$, $p=0.06$, $\eta_p^2 = 0.12$), but not for DNP vs PLC ($F_{1,27}=1.61$, $p=0.22$, $\eta_p^2 = 0.06$,

$BF_{01}$=2.00). The beneficial effect of cue validity on RT was not modulated by drug ($F_{2,54}$=1.35, p=0.27, $\eta_p^2$ = 0.05, $BF_{01}$=3.33). The effect of drug condition on behavior was more subtle compared to the effects of cue validity. First, the main effect of drug condition on d' was not significant ($F_{2,54}$=2.80, p=0.07, $\eta_p^2$ = 0.09; *Figure 2A*), but planned pairwise rmANOVAs indicated that d' was improved by ATX, but not by DNP, compared to PLC (ATX: $F_{1,27}$=4.28, p=0.048, $\eta_p^2$ = 0.14; DNP: $F_{1,27}$=1.61, p=0.22, $\eta_p^2$ = 0.06, $BF_{01}$=1.58; *Figure 2A*). RTs were not affected by drug ($F_{2,54}$=1.33, p=0.27, $\eta_p^2$ = 0.05, $BF_{01}$=1.99; *Figure 2B*), and pairwise comparisons (vs. PLC) were also not significant for neither ATX nor DNP (ATX: $F_{1,27}$=1.39, p=0.25, $\eta_p^2$ = 0.05, $BF_{01}$=1.59; DNP: $F_{1,27}$=2.82, p=0.11, $\eta_p^2$ = 0.10, $BF_{01}$=0.95). Participants did not exhibit changes in response bias for the different condition (for the full results see *Figure 2—figure supplement 1*).

Perceptual sensitivity and RT are behavioral measures that only capture the final output of the decision process and are not directly informative with respect to the components underlying the formation of a decision. Computational models that are fitted to behavioral data fill this gap by decomposing choice behavior into constituent cognitive processes (*Forstmann et al., 2016*). Besides this, another advantage of computational modeling over analyses of choice behavior is that it integrates various dependent variables – that is, choice accuracy and response time distributions – resulting in more robust descriptions of the underlying data. Here, we aimed to quantify the effects of drug and cue validity on latent variables of the decision-making process, by fitting a drift diffusion model (DDM; *Ratcliff and McKoon, 2008*) to behavioral data (see Materials and methods). The DDM describes perceptual decisions as the accumulation of noisy evidence over time that is terminated when accumulated evidence crosses a certain decision threshold (*Figure 2C*). The basic version of the model consists of three variables: drift rate (reflecting the average rate of evidence accumulation; v), decision boundary separation (reflecting the amount of evidence required to commit to a decision; b), and non-decision time (the duration of non-decision processes, including sensory encoding and motor execution; t0). Based on previous literature, we hypothesized that cue validity would mainly affect sensory evidence accumulation reflected in drift rate (*Loughnane et al., 2016*; *van Vugt et al., 2019*), further suggested by the fact that valid cues resulted in better and faster performance on Gabor discrimination (*Figure 2A and B*). On the contrary, although perceptual sensitivity improved under ATX (vs. PLC), average RTs were not affected, suggesting that other parameters may also have been modulated by drug besides drift rate. So, to tease apart the effects of cue validity and drug condition, we fitted a model in which drift rate, boundary separation, and non-decision time were allowed to fluctuate with cue validity and drug. We included drift rate variability to account for between trial variance in drift rate. We fitted two hierarchical Bayesian regression models to capture within-subject effects of both drug and cue validity: one for ATX vs. PLC ('ATX model') and one for DNP vs. PLC ('DNP model'). Note that these regression models did not allow to include all three drug conditions in one model and therefore we applied a 2×2 factorial design (cue validity × drug) for both ATX and DNP separately. The Bayesian implementation of the DDM constrained single-subject parameter estimates based on group-level parameter distributions, thereby increasing robustness to outliers. Model fits are reported in *Figure 2—figure supplement 2*.

We indeed found that validly cued trials were associated with increased drift rates. This can be observed in *Figure 2D* by a clear divergence of the posterior probability distributions for the ATX model (in blue) and the DNP model (in green) from 0 (both models p<0.001; *Figure 2D*, 'cue validity effects'). We also observed increased drift rate under ATX (in blue), but not for DNP (ATX model: p=0.02, DNP model: p=0.09; *Figure 2D*, 'drug effects'). The means of the posterior distributions (an indication of effect size) for the effects of cue validity and ATX on drift rate were comparable (ATX model: $\bar{x}_{cue}$=0.17, $\bar{x}_{ATX}$=0.15), although the estimates of cue validity effects were more precise than those of ATX (ATX model: $s_{cue}$ = 0.03, $s_{ATX}$ = 0.07). Moreover, ATX increased the effects of cue validity on drift rate, but the effect size was small in comparison to the main effects of ATX and cue validity (ATX model: p=0.02; $\bar{x}_{interaction}$=0.04; *Figure 2D*, 'interaction effects'). DNP did not significantly modulate the effects of cue validity on drift rate, although the direction of this modulation was similar to ATX (DNP model: p=0.07; *Figure 2D*).

Although the rate of evidence accumulation was increased under ATX, the onset of this accumulation was delayed, indexed by an increase in non-decision time (ATX model: p=0.048, DNP model: p=0.09; *Figure 2E*). The effect of cue validity on non-decision time was not robust, when taking both models into account, although the effect was significant in the DNP model (ATX model: p=0.14, DNP

model: p=0.02; *Figure 2E*). We also did not observe any interaction effects between cue validity and drug on non-decision time (ATX model: p=0.11, DNP model: p=0.38; *Figure 2E*). Cue validity and drug did not affect decision bound separation, showing that response caution was not under the influence of neuromodulation or attention (*Figure 2—figure supplement 3*). We fitted two additional models (one for ATX vs. PLC and one for DNP vs. PLC) that allowed drift rate variability to also fluctuate with cue validity and drug, while keeping decision bound fixed. We used this model to test whether the effects of cue validity and ATX were indeed related to overall drift rate and not drift rate variability (*Murphy et al., 2014*). We observed no effects of cue validity or drug on drift rate variability showing that the overall rate, and not the consistency, of evidence accumulation was enhanced by ATX and cue validity (*Figure 2—figure supplement 4*).

In sum, the DDM results revealed robust effects of attention and catecholaminergic enhancement on decision-making at the algorithmic level. Valid cues and ATX both enhanced the rate of evidence accumulation toward a decision threshold and ATX also increased the effect of cue validity on drift rate, although this interaction effect was smaller than the respective main effects of ATX and cue validity. Cholinergic main and interaction effects were often in the same direction as for ATX, but not statistically reliable. These DDM results were echoed in the behavioral performance measures (especially d'), although less robustly, highlighting the benefits of combining several behavioral measures (accuracy and RT) into drift diffusion modeling.

## Neuromodulatory changes in evidence accumulation reflected in centro-parietal accumulation signal

Evidence accumulation (defined at the algorithmic level as drift rate) is thought to be reflected in neural activity in centro-parietal cortex (*Gold and Shadlen, 2007*). A recently identified positively trending signal over human centro-parietal regions, termed the centro-parietal positivity (CPP), has been shown to track task difficulty, RTs, and task performance during challenging perceptual decisions (*O'Connell et al., 2012*; *van Vugt et al., 2019*). The CPP furthermore scales independently of stimulus identity (unsigned component, identical for two or more stimulus categories), correlates with drift rate as derived from modeling behavior with DDMs, and as such is regarded as a neural reflection of the decision variable (i.e. accumulated evidence, *Figure 2C*; *Twomey et al., 2015*).

We plotted the CPP locked to the response in *Figure 3*. Note that we have transformed the EEG data to current source density (CSD) to sharpen the spatial sensitivity of our EEG results, which is suggested to be optimal for CPP analyses (*Kelly and O'Connell, 2013*; non-CSD-transformed topographic maps can be observed in *Figure 3—figure supplement 1*). In line with previous studies (*O'Connell et al., 2012*; *van Vugt et al., 2019*), the CPP predicted the accuracy and speed of perceptual decisions: CPP amplitude at the time of the response and its slope in the time-window starting –250 ms preceding the response were both higher for correct than erroneous decisions (amplitude: $F_{1,27}$=34.28, p<0.001, $\eta_p^2 = 0.56$; slope: $F_{1,27}$=33.20, p<0.001, $\eta_p^2 = 0.55$; *Figure 3A*), whereas decision speed was uniquely associated with CPP slope (amplitude: $F_{1,27}$=2.06, p=0.16, $\eta_p^2 = 0.07$, $BF_{01}$=1.60; slope: $F_{1,27}$=10.23, p=0.004, $\eta_p^2 = 0.27$; *Figure 3B*). Moreover, subjects with high drift rate (across all conditions, median split) showed both higher CPP amplitude and slope (one-way ANOVA; amplitude: $F_{1,26}$=6.83, p=0.01, $\eta_p^2 = 0.21$; slope: $F_{1,26}$=5.03, p=0.03, $\eta_p^2 = 0.16$; *Figure 3C*), indicating that the CPP indeed reflects accumulated evidence over time (*O'Connell et al., 2012*; *Twomey et al., 2015*).

After establishing the CPP as a reliable marker of evidence accumulation, we inspected its relation to neuromodulatory drive and cue validity (*Figure 3D*). Interestingly, drug condition affected the amplitude and slope of the CPP in a robust manner (main effects of drug, amplitude: $F_{2,54}$=11.26, p<0.001, $\eta_p^2 = 0.29$, *Figure 3E*; slope: $F_{2,54}$=4.31, p=0.02, $\eta_p^2 = 0.14$, *Figure 3F*). Specifically, CPP amplitude was heightened by ATX compared to PLC (amplitude: $F_{1,27}$=4.73, p=0.04, $\eta_p^2 = 0.15$; slope: $F_{1,27}$=0.47, p=0.50, $\eta_p^2 = 0.02$, $BF_{01}$=4.08), whereas DNP lowered CPP peak amplitude and decreased its slope compared to PLC (amplitude: $F_{1,27}$=8.71, p=0.01, $\eta_p^2 = 0.24$; slope: $F_{1,27}$=4.91, p=0.04, $\eta_p^2 = 0.15$). Trials on which target locations were validly cued were associated with an increased CPP slope and amplitude (amplitude: $F_{1,27}$=6.13, p=0.02, $\eta_p^2 = 0.19$, *Figure 3E*; slope: $F_{1,27}$=14.70, p<0.001, $\eta_p^2 = 0.35$, *Figure 3F*). Drug condition and cue validity shaped CPP slope and peak amplitude independently because no significant interactions were observed (amplitude: $F_{2,54}$=0.10, p=0.90, $\eta_p^2 = 0.00$, $BF_{01}$=8.53, *Figure 3E*; slope: $F_{2,54}$=0.55, p=0.58, $\eta_p^2 = 0.02$, $BF_{01}$=6.08, *Figure 3F*, Bayesian evidence was in favor of the null). Summarizing, we show that drug condition and cue validity both

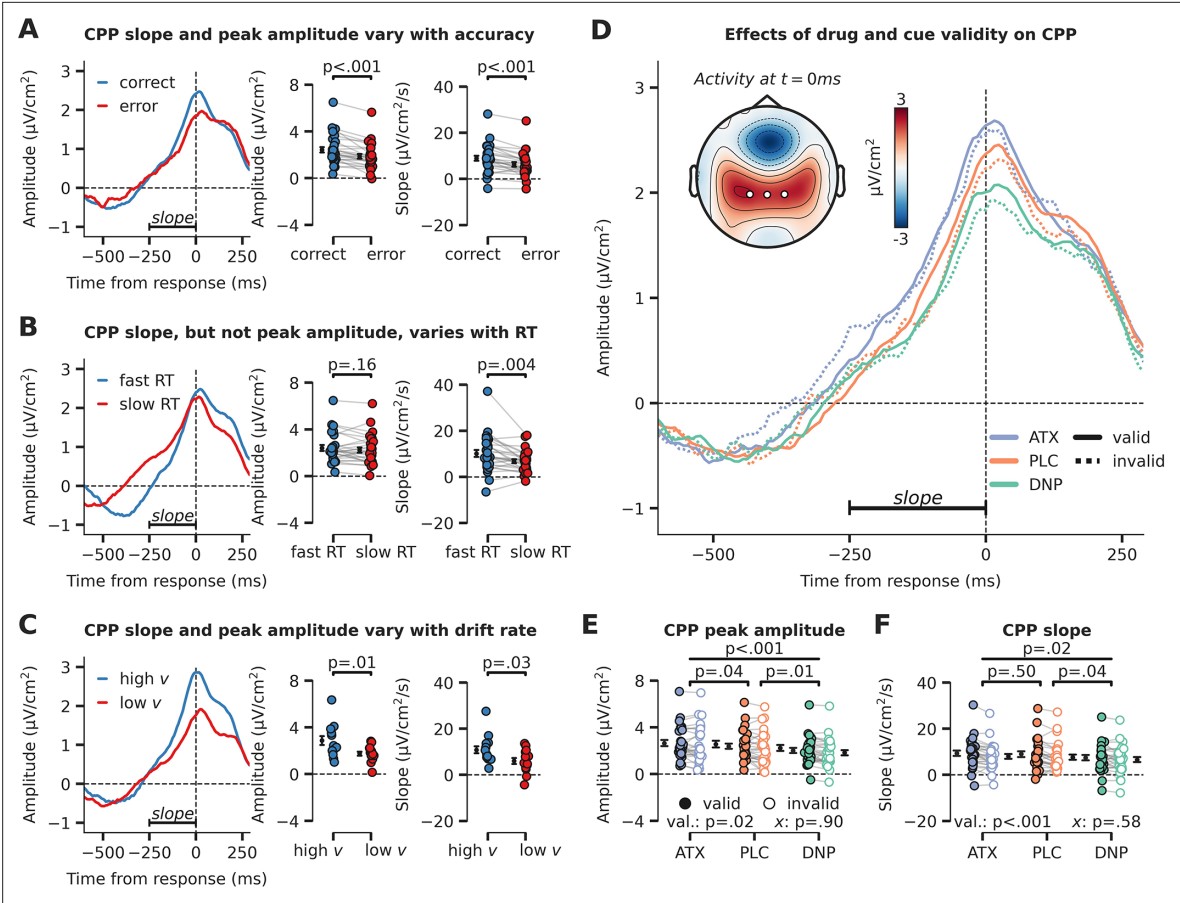

**Figure 3.** Evidence accumulation is affected by cue validity and drug, indexed by changes in centro-parietal positivity (CPP). (**A**) Response-locked CPP for correct and incorrect answers, (**B**) for trials with fast and slow reaction times (RTs) and (**C**) for participants with overall high drift rate and low drift rate. (**D**) Modulation of response-locked CPP by drug and cue validity. The horizontal black line indicates the time-window for which CPP slope was calculated (linear regression from –250 ms to 0 ms pre-response). The topographic map shows activation at the moment of the response, with white markers indicating the centro-parietal ROI used for the CPP analyses (channels CP1, CP2, CPz). (**E**) Peak CPP amplitude, separately for drug and cue validity. (**F**) CPP slope for all drug and cue validity conditions. Note that x demarks the p-value for the omnibus interaction effect between drug condition and cue validity and that val. (short for validity) refers to the factor cue validity. Error bars indicate SEM.

The online version of this article includes the following source data and figure supplement(s) for figure 3:

**Source data 1.** Source files for centro-parietal positivity (CPP) data and analyses.

**Figure supplement 1.** Centro-parietal positivity (CPP) topography without current source density (CSD) transformation.

**Figure supplement 1—source data 1.** Source files for centro-parietal positivity (CPP) topographical data (without current source density [CSD] transformation).

affected CPP peak amplitude (ATX, DNP and cue validity) and slope (DNP and cue validity) and interactions between these factors were not observed.

## Stimulus-locked signals over occipito-temporal sites precede evidence accumulation and are modulated by catecholaminergic stimulation and attention

Previous studies investigating the neural origins of perceptual decision-making during visual tasks have observed early EEG deflections at lateral occipito-temporal sites that reflect target selection prior to CPP build-up (*Loughnane et al., 2016*; *Newman et al., 2017*; *van Kempen et al., 2019*). These target-related signals in perceptual decision tasks are usually measured at electrode sites P7/P9 (left hemisphere) and P8/P10 (right hemisphere) and are referred to as the N2c (contralateral to the target) and N2i (ipsilateral to the target; *Loughnane et al., 2016*; *van Kempen et al., 2019*). Similar to the CPP, N2c/i amplitudes scale with RT and stimulus coherence and shifts in the peak latency of

the N2c have been shown to influence the onset of the CPP (*Loughnane et al., 2016*), suggesting that these signals may strongly modulate subsequent evidence accumulation signals.

To test whether the effects of increased neuromodulation and attention on the CPP were preceded by modulations of these occipito-temporal signals, we extracted stimulus-locked neural activity from these posterior lateral electrodes (based on literature; *Loughnane et al., 2016*; *Newman et al., 2017*; *Papaioannou and Luck, 2020*; *van Kempen et al., 2019*) as spatial regions of interest (left ROI: P7/P9; right ROI: P8/P10; *Figure 4A*). Next, we performed a cluster-corrected 2 × 2 × 3 factorial rmANOVA across time to test for effects of the cue (cued vs uncued), stimulus identity (target/noise), and drug (ATX/DNP/PLC), as well as possible interaction effects. To avoid potential confusion, the relationship between the factors cue and stimulus identity is further explicated in *Figure 4A*. We plotted the ERP traces for all 12 conditions in *Figure 4B* (stimulus-locked). To anticipate the results, we only observed main effects of each of the three factors and no interactions. In *Figure 4C*, we plotted the main effects of cue (left panel, stimulus cued vs. stimulus uncued), stimulus identity (middle panel, target vs. noise stimulus), and drug condition (right panel, all three drugs). For each main effect plot, data were averaged over the other experimental factors. Below we report the specific effects in more detail.

First, cued stimuli triggered a more positive (or less negative) ERP response than uncued stimuli already early on, specifically from 70 ms to 336 ms post-stimulus (two clusters: 70–125 ms, p=0.043; 141–336 ms, p<0.001; *Figure 4C* – left panel). The modulation of this activity (averaged across both temporal clusters: 70–336 ms) was similar for when the target stimulus was cued vs. when the noise stimulus was cued and therefore reflects an initial modulation of visual processing of any cued (attended) stimulus, irrespective of whether it was a target or not ($F_{1,27}$=0.08, p=0.77, $\eta_p^2 = 0.00$, $BF_{01}$=4.58; *Figure 4—figure supplement 1A*). In other words, there was no cue validity effect, which would have been reflected in an interaction between cue and stimulus identity. Moreover, the cue-related modulation of neural activity (cued vs. uncued) was not affected by drug condition ($F_{2,54}$=0.01, p=0.99, $\eta_p^2 = 0.00$, $BF_{01}$=8.21) and we also observed no three-way interaction between cue, stimulus, and drug in this time-window ($F_{2,54}$=0.57, p=0.57, $\eta_p^2 = 0.02$, $BF_{01}$=3.76; *Figure 4—figure supplement 1A*).

Second, the main effect of stimulus identity on occipito-temporal activity (difference between target vs. noise stimuli) occurred later in time, from 352 ms to 500 ms post-stimulus (p=0.01; *Figure 4C* – middle panel). This effect was not modulated by drug condition ($F_{2,54}$=0.74, p=0.48, $\eta_p^2 = 0.03$, $BF_{01}$=9.63) or cue ($F_{1,27}$=1.31, p=0.26, $\eta_p^2 = 0.05$, $BF_{01}$=2.95) and we also did not observe a significant three-way interaction ($F_{2,54}$=0.80, p=0.46, $\eta_p^2 = 0.03$, $BF_{01}$=4.46; *Figure 4—figure supplement 1B*).

Third, and finally, similar to the effect of stimulus identity, drug condition affected ERP amplitudes relatively late in time (406–586 ms; p=0.004; *Figure 4C* – right panel). Specifically, ATX showed a trend toward lower ERP amplitudes compared to PLC ($F_{1,27}$=3.56, p=0.07, $\eta_p^2 = 0.12$; *Figure 4D* – right panel), whereas DNP increased amplitudes over occipito-temporal regions in this time-window ($F_{1,27}$=4.86, p=0.04, $\eta_p^2 = 0.15$). We did not observe any significant interaction effects with drug condition and cue on ERP amplitude in this time-window ($F_{2,54}$=0.37, p=0.69, $\eta_p^2 = 0.01$, $BF_{01}$=6.43).

## Neuromodulatory effects on preparatory attention

The predictive cues in our task allowed participants to anticipate targets appearing at certain locations, by covertly shifting their locus of attention. Previous work has identified a neural marker for attentional orienting in response to a visual cue, that is an early negative amplitude deflection over contralateral regions (early directing attention negativity [EDAN]; *Murray et al., 2011*; *Praamstra and Kourtis, 2010*). Moreover, attentional shifts have been associated with changes in cortical excitability, indexed by lateralized alpha-band (8–12 Hz) power suppression (stronger alpha suppression at the contralateral side of the stimulus; *Capotosto et al., 2009*; *Händel et al., 2011*; *Jensen and Mazaheri, 2010*; *Thut et al., 2006*). Recent studies also suggests that neuromodulatory systems may be involved in regulating changes in global cortical excitability in relation to shifts in spatial attention (*Bauer et al., 2012*; *Dahl et al., 2022*). Here, we tested whether enhanced catecholaminergic and cholinergic levels modulated these electrophysiological markers of preparatory spatial attention. For the following EEG analyses, we used two symmetrical ROIs over occipital areas that are commonly used for analyses regarding preparatory attention (left ROI: O1, PO3, and PO7; right ROI: O2, PO4, and PO8; *Capotosto et al., 2009*; *Kelly et al., 2006*; *Thut et al., 2006*).

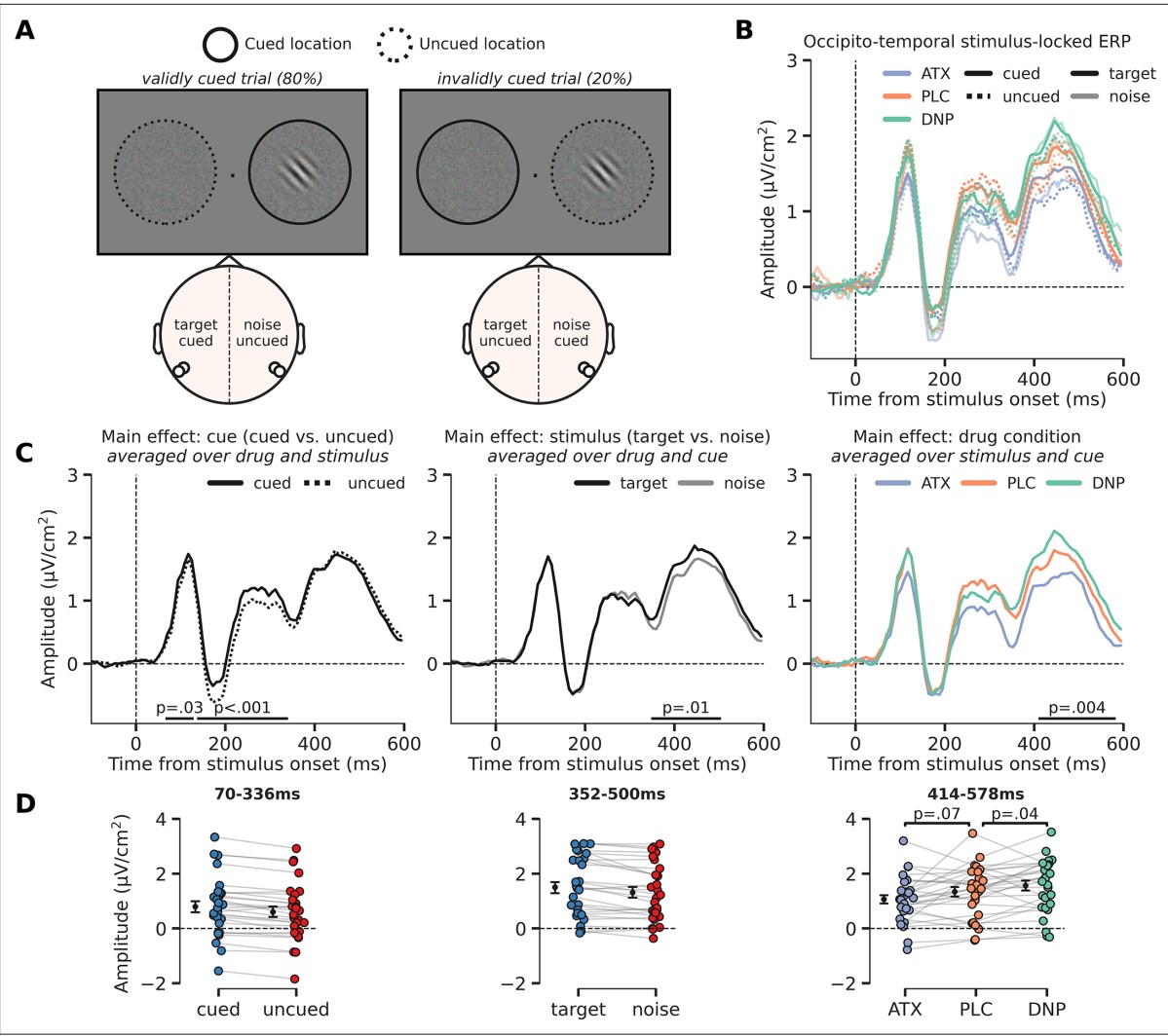

**Figure 4.** Modulation of perceptual processing by drug and attention, indexed by changes in occipito-temporal activity. (**A**) Schematic of the definition of ERP conditions, based on the cued location and the location of the target stimulus. The cue was presented before the stimulus and is not shown here, but the cued and uncued location are, respectively, illustrated by solid and dashed circles around the stimulus (not shown in the actual experiment). We extracted activity from bilateral occipito-temporal regions of interest (ROIs) (left ROI: P7/P9 and right ROI: P8/P10; see white markers in topographic map), separate for target stimuli (ROI contralateral to target) and noise stimuli (ROI ipsilateral to the target) and cued stimuli (ROI contralateral to cued location) and uncued stimuli (ROI ipsilateral to cued location). This was done for each drug condition. In the schematic we show two example scenarios for when the target stimulus appeared on the right side of fixation (it could also appear on the left side of fixation). In the first example, in the left panel, the target stimulus was cued (validly cued trial), in which case the left ROI shows activity related to the 'cued target' and the right ROI shows activity related to the 'uncued noise' stimulus. In the second example, in the right panel, the noise stimulus was cued (invalidly cued trial) in which case the right ROI shows activity related to the 'cued noise' stimulus and the left ROI shows activity related to the 'uncued target' stimulus. By defining ERP traces as such, we were able to investigate the effects of the spatial cue in isolation (i.e. cued vs. uncued, spatial attention effect) and in interaction with stimulus identity (i.e. cue validity effect). (**B**) Stimulus-locked ERPs (0 ms = stimulus presentation) over regions processing the target stimulus (i.e. contralateral to target stimulus, opaque line) and the noise stimulus (i.e. ipsilateral to target stimulus and/or contralateral to noise stimulus, transparent line). These traces are further split up depending on cue condition, meaning whether the stimulus was cued (i.e. contralateral to the cue, continuous line) or uncued (i.e. ipsilateral to the cue, dashed line). Note that the traces of *validly cued* trials can be seen in both cued target (solid, opaque lines) and uncued noise trials (dashed, transparent lines). Finally, the traces are plotted for the different drug conditions. (**C**) The main effects of cue (cued vs uncued, left panel), stimulus identity (target vs. noise, middle panel), and drug condition (ATC/donepezil [DNP]/placebo [PLC], right panel). (**D**) Data extracted from these cluster for each of the conditions of the main effect. Post hoc two-sided t-test were performed to test the respective effects of atomoxetine (ATX) and DNP vs. PLC (right panel). Error bars indicate SEM.

The online version of this article includes the following source data and figure supplement(s) for figure 4:

**Source data 1.** Source files for stimulus-locked occipital ERP data.

**Figure supplement 1.** Post-hoc analyses of interactions with stimulus-locked cue-effect and stimulus effect.

**Figure supplement 1—source data 1.** Source files for additional stimulus-locked occipital ERP analyses.

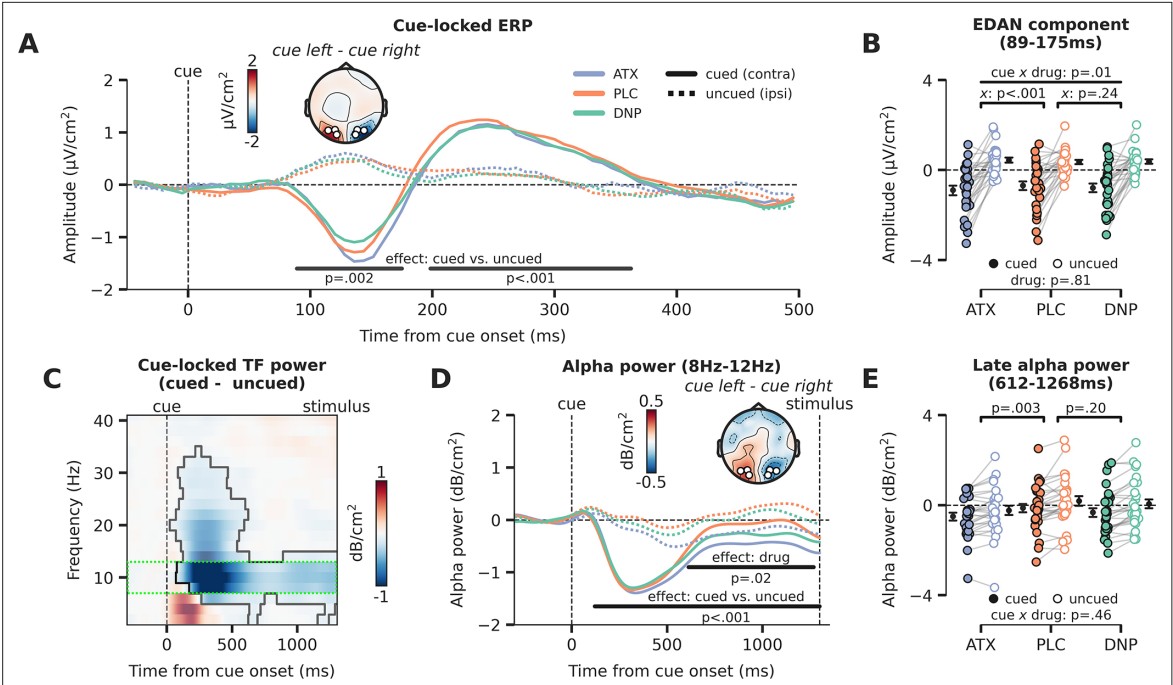

**Figure 5.** Attentional and catecholaminergic modulation of prestimulus cortical activity. (**A**) Cue-locked ERP (μV/cm²) over occipital regions reflecting the cued location (hemisphere contralateral to the cued direction) and reflecting the uncued location (hemisphere ipsilateral to the cued direction), split up for the three drug conditions. Black horizontal bars show repeated measures ANOVA (rmANOVA) main effects of cue, cued (contra) vs. uncued (ipsi), cluster corrected for multiple comparisons. The topographic map shows the contrast between a cue to the left vs. a cue to the right (time-window: 89–175 ms, early cluster in which we observed the lateralization effect). White markers indicate the regions of interest (ROIs) used for all cue-locked analyses. (**B**) Average activity within the early temporal cluster for the hemisphere associated with the cued (contra) vs uncued (ipsi) location as a factor of drug, showing lateralized effects. The p-value of the interaction effect between cue and drug (pairwise vs. placebo [PLC]) is demarked with x. (**C**) The difference in time-frequency (TF) power (dB/cm²) for cued (contra) vs uncued (ipsi) locations (lateralization effect). The significant TF cluster (highlighted by the solid black line) is derived from cluster-based permutation testing controlling for multiple comparisons. The green dotted box indicates the alpha-band TF ROI that is used to compute panel D. (**D**) Cue-locked prestimulus alpha power (dB/cm²) over occipital regions associated with the cued (contra) and uncued (ipsi) location, split up for the three drug conditions. Black horizontal bars show rmANOVA main effects for cue (contra vs. ipsi, lateralization effect) and drug. The topographic map shows the contrast between when a cue to the left vs. a cue to the right was presented (time-window: 659–1253 ms, cluster in which we observed a main effect of drug). (**E**) Average alpha power within the late temporal cluster in which drug effects on alpha power were observed, split up for cue and drug conditions. The p-values on top reflect the main effects between atomoxetine (ATX)/donepezil (DNP) and placebo (PLC) and the p-value at the bottom demarked with x depicts the omnibus interaction effect between drug condition and cue. Error bars indicate SEM.

The online version of this article includes the following source data and figure supplement(s) for figure 5:

**Source data 1.** Source files for cue-locked ERP and time-frequency data.

**Figure supplement 1.** Late cue-locked ERP effects and relation between prestimulus alpha power and behavioral cue validity effects.

**Figure supplement 1—source data 1.** Source files for late cue-locked ERP cluster and alpha data.

In the next set of analyses, we will inspect prestimulus activity for the three different drug conditions, locked to the attentional cue. Therefore, there is no factor stimulus identity, but the logic defining the cue conditions is the same as illustrated in *Figure 4A*. The factor cue again had two levels and referred to the hemisphere associated with the cued location (the hemisphere contralateral to the cued direction) and the hemisphere associated with the uncued location (the hemisphere ipsilateral to the cued direction). For simplicity, we will refer to these factor levels as the cued (contra) and the uncued (ipsi) location from here on. We plotted the cue-locked ERPs for all six conditions in *Figure 5A* (cue × drug). Neural activity for the hemisphere associated with the cued location (contra) was clearly different than for the uncued location (ipsi) in two time-windows (89–175 ms, p=0.002; 198–363 ms, p<0.001). This analysis thus revealed a strong lateralization of neural activity due to the shift of attention toward the cued location. Interestingly, the early lateralized component, resembling the EDAN, was modulated by drug (interaction: drug × cue, $F_{2,54}$=5.05, p=0.01, $\eta_p^2$ = 0.16; *Figure 5B*) and this interaction was

driven by ATX (ATX vs. PLC: $F_{1,27}$=23.95, p<0.001, $\eta_p^2 = 0.47$; DNP vs. PLC: $F_{1,27}$=1.47, p=0.24, $\eta_p^2 = 0.05$, $BF_{01}$=2.12). Cue-related lateralized activity in the later cluster showed a trending effect of overall drug modulation ($F_{2,54}$=3.12, p=0.05, $\eta_p^2 = 0.10$; *Figure 5—figure supplement 1A*), but pairwise post hoc tests showed no reliable differences for ATX/DNP vs. PLC when tested separately (ATX: $F_{1,27}$=1.44, p=0.24, $\eta_p^2 = 0.05$, $BF_{01}$=2.31; DNP: $F_{1,27}$=1.32, p=0.26, $\eta_p^2 = 0.05$, $BF_{01}$=2.61).

Next, we focused on alpha-band dynamics from the onset of the attentional cue leading up to stimulus presentation (1300 ms post-cue). Replicating previous work, the time-frequency (TF) spectrum plotted in *Figure 5C* shows that attentional cues resulted in a stronger suppression of alpha-band power for the hemisphere associated with the cued location (contra) compared to the uncued (ipsi) location. To relate these effects to drug condition, we plotted alpha power (8–12 Hz) for the hemisphere associated with the cued and uncued locations for all three drug conditions separately (*Figure 5D*). Stronger alpha-band desynchronization for cued vs uncued locations was observed 120 ms from cue onset until target stimulus onset (p<0.001; *Figure 5D*). Moreover, there was a main effect of drug, revealing that total alpha power over both the hemisphere associated with the cued location and the hemisphere associated with the uncued location was modulated by drug from 659 ms to 1253 ms after cue onset (p=0.02; *Figure 5D*). Again, this main effect of drug was driven by catecholaminergic activity (ATX: $F_{1,27}$=10.42, p=0.003, $\eta_p^2 = 0.28$; DNP: $F_{1,27}$=1.73, p=0.20, $\eta_p^2 = 0.06$, $BF_{01}$=1.30; *Figure 5E*). Drug condition did not modulate the effect of cue on prestimulus alpha power ($F_{2,54}$=10.42, p=0.46, $\eta_p^2 = 0.03$, $BF_{01}$=4.99; *Figure 5E*). This latter result shows that ATX reduced overall alpha-band power for both cued (contra) and uncued (ipsi) locations, with respect to PLC, so not in a lateralized manner. The effect of the spatial cue on prestimulus alpha-band power lateralization in this late time-window was related to the cue effect on perceptual sensitivity and this relation was not different between the drug sessions (see *Figure 5—figure supplement 1B* for details).

In sum, replicating a wealth of previous attention studies, both occipital ERPs (EDAN) and occipital alpha-band suppression were affected by shifting the locus of spatial attention, as reflected in clear neural differences between activity measured over the hemisphere associated with cued (contra) vs. the uncued (ipsi) location. Interestingly, ATX increased the lateralized cue-locked ERP component (EDAN) by enhancing the difference between cued (contra) and uncued (ipsi) locations, suggesting enhanced attentional orientation in a hemisphere-specific manner (*Figure 5A*). However, in contrast, prestimulus occipital alpha power, an index of cortical excitability (*Capotosto et al., 2009*; *Händel et al., 2011*; *Jensen and Mazaheri, 2010*), was bilaterally decreased under ATX, and did not relate to the attentional cue, suggesting instead that enhanced catecholaminergic levels regulate cortical excitability in a spatially non-specific and global manner (*Figure 5C*).

## Discussion

In this study, we used a 3×2 factorial design in which we manipulated neuromodulator levels (through pharmacology) and spatial attention, allowing us to causally demonstrate their separate, shared, and interactive effects on perceptual decision-making in healthy human participants. By administering ATX and DNP, we were further able to tease apart the roles of cholinergic and catecholaminergic neuromodulatory systems respectively, in relation to, and in interaction with, spatial attention.

### Spatial attention and catecholaminergic neuromodulation jointly shape perceptual decision-making

Computational modeling of choice behavior showed that both the allocation of attention to target stimuli (rotated Gabor patches) and increased catecholaminergic levels enhanced the rate of sensory evidence accumulation (drift rate, *Figure 2D*), resulting in improved perceptual sensitivity (effect was stronger for attention, *Figure 2A*). In other words, participants were better able to tell whether a presented Gabor stimulus was oriented clockwise or counterclockwise when Gabors were attended (validly cued vs. invalidly cued) and when ATX was received (vs. PLC). These effects were reflected in a well-known neural marker of sensory evidence accumulation measured over parietal cortex (CPP, *Figure 3D*), as elevated catecholaminergic levels increased CPP peak amplitude and attention increased both its slope and its peak amplitude. Although the effects of attention and catecholaminergic neuromodulation on algorithmic and neural markers of sensory evidence accumulation were ostensibly similar, additional EEG analyses revealed that attention and catecholamines differently

modulated activity over occipito-temporal regions. First, we observed that the effects of attention and catecholaminergic neuromodulation on stimulus-locked (Gabor) neural activity over occipito-temporal electrodes occurred at different latencies. Attention modulated sensory processing much earlier in time (starting ~70 ms after stimulus presentation) than the catecholaminergic intervention (starting ~400 after stimulus presentation; *Figure 4A*). Second, both spatial attention and catecholamines modulated alpha-band desynchronization, a marker of cortical excitability (*Figure 5D*), but attention did so unilaterally, as reflected in stronger alpha-band desynchronization in the hemisphere opposite to the locus of attention, whereas catecholaminergic modulation of alpha-band desynchronization was bilateral (no hemispherical specificity nor interaction with attentional cue). Both findings are in line with recent work showing independent modulations of spatial attention and arousal on neuronal activity in mouse V1 (*Kanamori and Mrsic-Flogel, 2022*). Thus, prestimulus cortical excitability (alpha power) and stimulus-locked activity related to the processing of visual input were differentially modulated by spatial attention and catecholaminergic neuromodulation.

We did observe two striking interaction effects between the catecholaminergic system and spatial attention. First, effects of attention on drift rate were increased under catecholaminergic enhancement (*Figure 2D*). Although this interaction effect was not reflected in CPP slope/peak amplitude, this does suggest that catecholamines and spatial attention might together shape sensory evidence accumulation in a multiplicative manner. Second, the amplitude of the cue-locked early lateralized ERP component (resembling the EDAN) was increased under ATX as compared to PLC. The underlying neural processes driving the EDAN ERP, as well as its associated functions, have been a topic of debate. Some have argued that the EDAN reflects early attentional orienting (*Praamstra and Kourtis, 2010*) but others have claimed it is mere a visually evoked response and reflects visual processing of the cue (*van Velzen and Eimer, 2003*). Thus, whether this effect reflects a modulation of ATX on early attentional processes or rather a modulation of early visual responses to sensory input in general is a matter for future experimentation.

Taken together, spatial attention and neuromodulation showed similar, unique, and interactive effects on neural activity underlying different stages of the perceptual decision-making process. This intriguing relation between spatial attention and neuromodulation advances our understanding about how these systems jointly shape perceptual decision-making but also indicates the necessity for further exploration of the complex interactions between these systems.

## Tonic versus phasic neuromodulation in relation to attentional processes

Most recent work associates phasic (event-related) arousal with selective attention (for reviews see: *Dahl et al., 2022*; *Thiele and Bellgrove, 2018*). For example, cue detection in visual tasks is known to be related to cholinergic transients occurring after cue onset (*Howe et al., 2017*; *Parikh et al., 2007*). Contrarily, in our work we aimed to investigate the effects of increased baseline levels of neuromodulation by suppressing the reuptake of catecholamines and the breakdown of ACh throughout cortex and subcortical structures. We are not the first to investigate the role of tonic neuromodulation in spatial attention, however. Previously, *Bauer et al., 2012*, showed that the effect of (non-probabilistic) spatial cues on prestimulus alpha power lateralization was increased under a cholinergic agonist (physostigmine) compared to PLC. We did not replicate this finding, neither for DNP (elevating cholinergic levels; *Figure 5E*) nor ATX (elevating catecholaminergic levels; *Figure 5E*). Whether this discrepancy in results is due to the difference in applied pharmaceuticals (DNP/ATX vs. physostigmine), the different nature of the spatial cue (probabilistic vs. not probabilistic), or something else, requires further investigation.

The respective roles of tonic and phasic neuromodulation in controlling (aspects of) attention – and behavior in general – is still unresolved (*de Gee et al., 2021*; *de Gee et al., 2014*; *de Gee et al., 2020*; *McGinley et al., 2015a*; *McGinley et al., 2015b*; *van Kempen et al., 2019*). One issue that complicates the matter is the difficulty to investigate causal effects of tonic neuromodulation in isolation of changes in phasic neuromodulation, mostly because phasic and tonic activity are thought to be anti-correlated, with lower phasic responses during high baseline activity and vice versa (*Aston-Jones and Cohen, 2005*; *de Gee et al., 2020*; *Knapen et al., 2016*). As such, pharmacologically elevating tonic neuromodulator levels may have resulted in changes in phasic neuromodulatory responses as well. Concurrent and systematic modulations of tonic (e.g. with pharmacology) and phasic (e.g. with

accessory stimuli; **Bruel et al., 2022**; **Tona et al., 2016**) neuromodulator activity may be necessary to disentangle the respective and interactive effects of tonic and phasic neuromodulator activity on human perceptual decision-making.

## Relating the algorithmic and neural implementations of evidence accumulation

We reported attentional and neuromodulatory effects on algorithmic (DDM, *Figure 2*) and neural (CPP, *Figure 3*) markers of sensory evidence accumulation. Recent work has started to investigate the association of these two descriptors of the accumulation process, aiming to uncover whether neural activity over centro-parietal regions reflects evidence accumulation, as proposed by computational accumulation-to-threshold models (**Kelly and O'Connell, 2015**; **O'Connell et al., 2018**; **O'Connell and Kelly, 2021**; **Twomey et al., 2015**). Currently, the CPP is often thought to reflect the decision variable, that is the (unsigned) evidence for a decision (**Twomey et al., 2015**), and consequently its slope should correspond with drift rate, whereas its amplitude at any time should correspond with the so far accumulated evidence. As – computationally – the decision is reached when evidence crosses a decision bound (the threshold), it may be argued that the peak amplitude of the CPP (roughly) corresponds with the decision boundary. This seems to contradict our observation that (1) ATX modulated drift rate, but not CPP slope and (2) ATX did not modulate boundary separation, but did modulate CPP peak. Note, however, that previous studies using pharmacology or pupil-linked indexes of (catecholaminergic) neuromodulation have also demonstrated effects on both CPP peak (**van Kempen et al., 2019**) and CPP slope (**Loughnane et al., 2019**).

Here, we demonstrated that response accuracy and response speed are differentially represented in the CPP, with correct vs. erroneous responses resulting in a higher slope and peak amplitude, whereas fast vs. slow responses are only associated with increased slopes (*Figure 3A and B*). Speculatively, the specific effect of any (pharmacological) manipulation on the CPP may depend on task setting. For example, *Loughnane et al., 2019*, used a visual task on which participants did not make many errors (hit rate >98%, no false alarms), whereas we applied a task in which participants regularly made errors (roughly 25% of all trials). Possibly, the effects of ATX from *Loughnane et al., 2019*, in terms of behavior (RT effect, not accuracy/d') and CPP feature (slope effect, not peak) may therefore have been different from the effects of ATX we observed on behavior (d' effect, not RT) and CPP feature (peak effect, not slope). Regardless, when we compared subjects with high and low drift rates (*Figure 3C*), we observed that both CPP slope and CPP peak were increased for the high vs. low drift group (independent of the drug or attentional manipulation). This indicates that both CPP slope and CPP peak were associated with drift rate from the DDM. Clearly, more work is needed to fully understand how evidence accumulation unfolds in neural systems, which could consequently inform future behavioral models of evidence accumulation as well.

## Attentional and neuromodulatory effects on sensory processing or decision-making?

Fluctuations in arousal states have been both associated with alterations in sensory representations (**Podvalny et al., 2021**; **Vinck et al., 2015**; **Warren et al., 2016**) and modulations of response criterion or other latent variables of the decision process (**de Gee et al., 2017**; **de Gee et al., 2014**; **de Gee et al., 2020**). Similarly, human work on alpha-band dynamics – often used as a proxy for attentional state – has not yet been able to arbitrate between the hypotheses that decreased (unlateralized) alpha-band power may reflect an overall increase in baseline excitability (resulting in amplified responses to both signal and noise) or whether baseline excitability may decrease/increase the trial-by-trial precision of neural responses (**Samaha et al., 2020**). The first case would result in alterations in response criterion (i.e. decision-making), whereas the latter case would lead to less/more accurate perceptual performance and sensitivity (i.e. likely related to perception) (**Busch et al., 2009**; **Iemi and Busch, 2018**; **Samaha et al., 2017**; **Samaha et al., 2020**; **Vanrullen et al., 2011**; **Zhou et al., 2021**). Here, we provide evidence that cholinergic and catecholaminergic neuromodulation seems to affect stimulus-locked neural responses relatively late in time (close to the execution of the motor response), whereas the effects of attention were observable much earlier. This corroborates earlier findings showing, respectively, late and early effects of neuromodulation and attention on cortical activity (**Alilović et al., 2019**; **Baumgartner et al., 2018**; **Gelbard-Sagiv et al., 2018**; **Hillyard and**

*Anllo-Vento, 1998*; *Loughnane et al., 2019*; *Murphy et al., 2011*; *Nuiten et al., 2021*; *Poort et al., 2012*). Although, based on the timing of these effects, it might be tempting to conclude that attention primarily modulates (early) perceptual processes and neuromodulation (late) decisional processes, this conclusion is at present premature. Our study was not set up to dissociate perceptual from decisional effects, as these are hard to disentangle in practice (see also *Sánchez-Fuenzalida et al., 2022*), and cannot be isolated based on timing alone (*Alilović et al., 2023*). However, this study highlights that various cortical states – for example attention and arousal – may distinctly affect stimulus-locked cortical activity and therefore need to be considered jointly when addressing the effect of (prestimulus) cortical state variations on perceptual decision-making (*Podvalny et al., 2021*; *Waschke et al., 2019*).

## Elevated cholinergic levels modulate neural activity in absence of bodily and behavioral effects

In line with previous non-clinical studies that report physiological responses of 5 mg DNP (*Pfeffer et al., 2018*; *Pfeffer et al., 2021*), we did not observe consistent subjective or physiological effects of DNP. However, even in the absence of such autonomic markers, 5 mg DNP has been shown to induce behavior and neural activity, related to visual perception (*Boucart et al., 2015*; *Gratton et al., 2017*; *Pfeffer et al., 2018*; *Pfeffer et al., 2021*; *Silver et al., 2008*). Although DNP did not significantly affect perceptual sensitivity and/or drift rate, we did observe a set of consistent neural effects, generally reflected in neural responses in the opposite direction as for ATX. This was the case for the CPP peak and slope (*Figure 3D and F*) as well as for occipito-temporal signals related to stimulus and cue processing (*Figure 4A/C*, *Figure 5A and B*). Corresponding with our observation that DNP and ATX effects were mirrored, a recent study using DNP and ATX in combination with whole-brain fMRI revealed that these neuromodulator systems have opposite effects on cortex-wide connectivity patterns, even though both systems enhanced neural gain (*Pfeffer et al., 2021*). More specifically, catecholamines enhanced and ACh suppressed cortex-wide auto-correlation of activity (*Pfeffer et al., 2021*), suggested to reflect intracortical signaling by lateral and feedback processes (*Hasselmo and Sarter, 2011*; *Silver et al., 2008*). Furthermore, high cholinergic levels are associated with enhanced afferent input to cortex (feedforward) but decreased feedback drive (*Hasselmo and Sarter, 2011*). Because both evidence accumulation processes and target selection processes strongly rely on cortical feedback (*Boehler et al., 2009*; *Dehaene et al., 2011*; *Donohue et al., 2020*; *Lamme and Roelfsema, 2000*; *Murphy et al., 2021*; *Pereira et al., 2022*; *Pitts et al., 2014*; *Seijdel et al., 2021*; *Theeuwes, 2010*; *Wang, 2008*), the mirrored (compared to ATX) effects of DNP on the neural markers of these processes may have been caused by interrupted feedback processing. Furthermore, although previous work has implicated the cholinergic system in prestimulus alterations in neural excitability (*Bauer et al., 2012*), indexed by alpha power desynchronization, we did not find evidence for this (*Figure 5E*), possibly because *Bauer et al., 2012*, used physostigmine, a more potent cholinergic esterase inhibitor than DNP (*Ogura et al., 2000*). Future work applying various stimulants and suppressors of cholinergic activity, administered in different dosages, is necessary to firmly establish the role of the cholinergic system in controlling cortical functioning and spatial attention.

## Limitations of the current study

Although the effects of the attentional manipulation were generally strong and robust, the statistical reliability of the effects of the pharmacological manipulation was more modest for some comparisons. This may partly be explained by high inter-individual variability in responses to pharmaceutical agents. For example, initial levels of catecholamines may modulate the effect of catecholaminergic stimulants on task performance, as task performance is supposed to be optimal at intermediate levels of catecholaminergic neuromodulation (*Cools and D'Esposito, 2011*). While acknowledging this, we would like to highlight that many of the observed effects of ATX were in the expected direction and in line with previous work. First, pharmacologically enhancing catecholaminergic levels have previously been shown to increase perceptual sensitivity (d') (*Gelbard-Sagiv et al., 2018*), a finding that we have replicated here. Second, methylphenidate, a pharmaceutical agent that elevates catecholaminergic levels as well, has been shown to increase drift rate as derived from drift diffusion modeling on visual tasks (*Beste et al., 2018*) in line with our ATX observations. Third, a previous study using ATX to elevate catecholaminergic levels observed that ATX increased CPP slope (*Loughnane et al., 2019*). Although

in our case ATX increased the CPP peak and not its slope, this provide causal evidence that centro-parietal ERP signals related to sensory evidence accumulation are modulated by the catecholaminergic system (*Nieuwenhuis et al., 2005*). Fourth, we observed that elevated levels of catecholamines affected stimulus-driven occipital activity relatively late in time and close to the behavioral response, which resonates with previous observations (*Gelbard-Sagiv et al., 2018*). Finally, ATX had robust effects on physiological responses (heart rate, blood pressure, pupil size), cue-locked ERP signals, and oscillatory power dynamics in the alpha-band, leading up to stimulus presentation. We concur, however, that more work is needed to firmly establish how (various forms of) attention and catecholaminergic neuromodulation affect perceptual decision-making.

## Conclusion

Summarizing, we show that elevated catecholaminergic levels and spatial attention jointly shape perceptual decision-making. Similar, unique, and interactive effects of attention and catecholaminergic neuromodulation on behavioral, algorithmic, and neural markers of decision-making were observed. Together, this study reveals an intricate, complex, and yet to be further explored, relationship between the attentional and catecholaminergic systems.

# Materials and methods

## Participants

For this experiment 30 male, Dutch-speaking, human participants were recruited from the online research environment of the University of Amsterdam. All participants were aged between 18 and 30. Given the pharmacological nature of this experiment, participants were only included after passing extensive physical and mental screening (see Screening procedure). This study was approved by the Medical Ethical Committee of the Amsterdam Medical Centre (AMC) and the local ethics committee of the University of Amsterdam. Written informed consent was obtained from all participants after explanation of the experimental protocol. Two participants decided to withdraw from the experiment after having performed the first experimental session. The data from these participants are not included in this work, resulting in N=28. Participants received monetary compensation for participation in this experiment.

## Screening procedure

Following registration, candidate participants were contacted via e-mail to inform them of the inclusion criteria, exact procedures, and possible risks. In addition, candidate participants were provided with the credentials of an independent physician that was available for questions and further explanation. The participants were contacted after a deliberation period of 7 days to invite them for a pre-intake via telephone. During this pre-intake, it was ensured that the candidate participant indeed met all inclusion and exclusion criteria. If so, participants were invited for an intake session at the research facility of the University of Amsterdam. During this in-person intake, the experimental protocol, including the following physical and mental assessment, was explained in detail after which candidate participants provided written consent. The intake further consisted of a set of physiological measures – body mass index, heart rate, blood pressure, and an electrocardiogram – and a psychiatric questionnaire to assess mental health. The data from the intake were assessed by a physician, who subsequently decided whether a candidate participant was eligible for participation. Lastly, participants performed the staircasing procedure for the behavioral task (see Staircasing procedure).

## Drug administration

The study was conducted using a randomized, double-blind crossover design. ATX (40 mg), DNP (5 mg), and PLC were administered in different experimental sessions, separated by at least 7 days. The order of drug administration was counterbalanced between participants. Experimental days started at 09:00 and ended at 16:00. ATX (~2 hr) and DNP (~4 hr) reach peak plasma levels at different latencies, therefore these pharmaceuticals were administered at different times prior to the onset of the behavioral tasks (*Figure 1A*). To ensure the double-blind design, participants were required to orally ingest a pill 4 hr prior to the onset of the first behavioral task, which could contain either DNP or a PLC, and a second pill 2 hr prior to onset of the first behavioral task, which could contain either ATX

or a PLC. Thus, participants received one PLC and a working pharmaceutical or two PLCs on every experimental session.

The pharmaceuticals used in this study were chosen based on their pharmacokinetic and pharmacodynamic properties, the relatively limited side-effects, and prior use of these pharmaceuticals in other studies in the cognitive sciences (*Boucart et al., 2015*; *Pfeffer et al., 2018*; *Pfeffer et al., 2021*; *Warren et al., 2016*). ATX is a relatively selective NA reuptake inhibitor, which inhibits the presynaptic NA reuptake transporter, thereby resulting in increased NA and dopamine levels in the synaptic cleft (*Simpson and Plosker, 2004*). The half-life of ATX varies between 4.5 and 19 hr and peak plasma levels are reached ~2 hr after administration. DNP is a cholinesterase inhibitor, which impedes breaking down of ACh by cholinesterase thereby resulting in increased ACh levels in the synaptic cleft (*Tiseo et al., 1998*). The elimination half-life of DNP is 70 hr and peak plasma levels are reached after ~4 hr.

Minor side-effects were reported by our participants, including a mild headache (1 DNP session, 2 ATX session) and mild nausea (1 DNP session, 4 ATX sessions). Other side-effects, such as fatigue and tenseness, were equally present across all drug sessions including PLC.

## Design and procedures

### Experimental setting

Participants performed several tasks during a single experimental day. The first task was an auditory discrimination/detection task, lasting 2 hr, that was executed directly after the administration of the first pill. The data of that experiment fall outside the scope of this paper. An EEG apparatus was connected 3.5 hr after ingestion of the first pill, ensuring that participants could start the main experiment precisely 4 hr after ingestion of the first pill, supposedly when blood concentration levels for both ATX and DNP were peaking. During this main part of the experiment participants performed five different computerized visual perception tasks, out of which we will only discuss the spatial attention task. The order of these behavioral tasks was counterbalanced between participants but was maintained over sessions. Participants were seated in a darkened, sound isolated room, resting their chins on a head-mount located 80 cm from a 69×39 cm screen (frequency: 60 Hz, resolution: 1920×1080). The main task and staircase procedure were programmed in Python 2.7 using PsychoPy (*Peirce, 2007*) and in-house scripts.

### Cued orientation discrimination paradigm

Participants performed a task that was adapted from the Posner cueing paradigm (*Posner, 1980*). For the task, participants were asked to report the orientation of Gabor patches as being clockwise (45°) or counterclockwise (–45°), by pressing left ('s' on a keyboard, for counterclockwise) or right ('k', for clockwise) with their left or right index finger, respectively. We did not specifically instruct participants to respond as quickly or accurately as possible. The Gabor patches were presented for 200 ms on either the left or right side of the screen (center at 8°, radius 5.5°, spatial frequency 1.365 cycles/degree). Simultaneously, circular patches containing dynamic noise were presented bilaterally (center at 8°, radius 6.5°). Prior to stimulus presentation, a visual cue was presented for 300 ms near the fixation mark (horizontal dash, center at 0.33°). This cue was predictive of the stimulus location with a cue validity of 80%, meaning that the cue matched the stimulus location in 80% of the trials. Participants were instructed to use the cue, by covertly shifting attention (i.e. without moving eyes) toward the cued location. In 20% of trials the cue was invalid and the stimulus would appear on the opposite side. After cue offset there was a 1000 ms interval until stimulus onset. The response window started concurrently with stimulus presentation and lasted 1400 ms. If participants did not respond during this window they would receive visual feedback informing them of their slow response (in Dutch: 'te laat', which translates into: 'too late'). A variable inter-trial interval (ITI) of (250–350 ms) started directly after a response or the end of the response window. Stimulus locations, stimulus orientation, and cue direction were all balanced to occur on 50% of the trials (i.e. 50% counterclockwise, 50% left location, 50% cue to the right, etc.). The task was divided in two blocks of 280 trials, which in turn were subdivided in shorter blocks of 70 trials, in between which the participants could rest.

To ensure participants did not shift their gaze toward either the left or right, we measured gaze position via eyetracking (see Eyetracking). Trials only commenced when participants' gaze was at fixation (cutoff: 1.5°) and whenever participants lost fixation or blinked during a trial, the fixation mark

would turn white and the data for the trial would be marked as faulty. At the end of each block, a new trial was presented for every trial that was terminated because of blinks or lost fixation. In total, participants performed 560 trials of this task without losing fixation or blinking. After 280 trials on which fixation was not lost, the eyetracker was recalibrated (see Eyetracking).

## Staircasing procedure

Participants performed a staircasing procedure during their intake session. The staircase task was almost identical to the primary task, but without predictive cues prior to stimulus onset. The stimulus properties, presentation time, and response window were the same. The ITI was prolonged to 1100–1300 ms. Participants received feedback on their performance on a trial-by-trial basis; the fixation dot turned green for correct answers and red for incorrect answers. An adaptation of the weighted up-down method proposed was used to staircase performance at 75% correct, by changing the opacity of the Gabor patch (*Kaernbach, 1991*). In short, opacity adjustment after erroneous responses were weighted differently than after correct responses, in a ratio of 3:1. The step size was 0.01 (opacity scale: 0–1, 1 is fully opaque), thus after errors the opacity would be increased by 0.01 and after correct answers it would be decreased by 0.01/3. The procedure was aborted after 50 behavioral reversals, that is changes in sequences from correct to error or vice versa. The output difficulty of the staircase procedure was calculated as the average opacity on reversal trials. In total, participants performed two blocks of this staircase procedure. The first block started at a high opacity, allowing the participants to become familiar with the stimulus. The second block started at the opacity obtained from the first block. After completing the staircasing procedure, task difficulty was fixed to be able to compare the effects of the pharmacological manipulation across all experimental sessions.

## Data acquisition and preprocessing

### Eyetracking

Gaze position and pupil size were measured with an EyeLink 1000 (SR Research, Canada) eyetracker during the experiment at 1000 Hz. Gaze position was moreover monitored online to ensure participants' gaze remained at or near fixation (within 1.5°, horizontal axis). A nine-point calibration was used at the start of each block. To minimize movement of the participant, we used a head-mount with chinrest. Throughout the experiment participants were instructed to move their heads as little as possible and to try to refrain from blinking during a trial. Pupil traces were bandpass filtered between 0.01 Hz and 10 Hz, blinks were linearly interpolated, and the effects of blinks and saccades on pupil diameter were removed via finite impulse-response deconvolution (*Knapen et al., 2016*).

### EEG acquisition, preprocessing, and TF decomposition

EEG data were recorded with a 64-channel BioSemi apparatus (BioSemi B.V., Amsterdam, The Netherlands), at 512 Hz. Vertical eye movements were recorded with electrodes located above and below the left eye, horizontal eye movements were recorded with electrodes located at the outer canthi of the left and the right eye. All EEG traces were re-referenced to the average of two electrodes located on the left and right earlobes. The data were high-pass filtered offline, with a cutoff frequency of 0.01 Hz. Next, bad channels were detected automatically via a random sample consensus algorithm (RANSAC), implemented in the Autoreject Python package (*Jas et al., 2017*), and subsequently interpolated via spline interpolation. Next, epochs were created by taking data from –2000 ms to 2000 ms around onset of stimulus presentation. To remove eyeblink artifacts, an independent component analysis (25 components) was performed on the epoched data and components that strongly correlated to vertical EOG data were excluded. On average 1.23 (sd: 0.47, maximum: 3) components were rejected per file. Remaining artifacts were automatically detected by using the same RANSAC algorithm as before but on epoched data. Bad segments were repaired via interpolation if the artifactual data was present in only a few channels, but if more channels were affected the epoch was removed from the EEG data. On average 3.82% (sd: 5.16, maximum: 24.18%) of all epochs were removed. Lastly, the scalp current density (CSD) was computed using the surface Laplacian to attenuate the effects of volume conductance (*Cohen, 2014*).

TF representations of EEG data were calculated from epochs that were extracted from a time-window –1600 ms to 0 ms prestimulus. TF power was obtained through convolution of Morlet wavelets

with epoched data from 2 Hz to 40 Hz in steps of 2 Hz. For each frequency, we defined the number of cycles as:

$$\text{number cycles (f)} = \frac{f}{2} \qquad (1)$$

Then we averaged TF data within cue direction (left/right) and cortical hemisphere (left/right) . Next, this average TF power, as well as single trial TF power, was normalized to a baseline of –250 ms to –50 ms pre-cue with a decibel (dB) transformation:

$$\text{power (dB)} = 10 * \log_{10} \frac{\text{signal}}{\text{baseline}} \qquad (2)$$

## Data analysis

All behavioral analyses were programmed in Python 3.7. Trials during which fixation was lost missed trials and trials with an RT larger than 1400 ms were excluded from all analyses. The remaining data was used for the main behavioral analyses, but for the alpha power binning analysis of *Figure 5— figure supplement 1B* only behavioral data belonging to unrejected EEG epochs were included to match single trial EEG to behavior. EEG analyses were performed with use of the Python package MNE (version 0.24.0; *Gramfort et al., 2013*).

### Statistical analysis of physiological data, side-effects, and self-reports

We obtained bodily (heart rate and blood pressure) and subjective (visual analogue scale [VAS]; *Bond and Lader, 1974*) measurements of arousal on three occasions during each session; before drug intake (baseline), prior to the onset of the first EEG task (after 4 hr), and at the end of each session (after 7 hr). We calculated mean arterial pressure (MAP) from systolic and diastolic blood pressure as:

$$\textit{mean arterial pressure} = \textit{diastolic BP} + \frac{1}{3} * \left( \textit{systolic BP} - \textit{diastolic BP} \right) \qquad (3)$$

VAS scores were calculated by measuring the location of marked answers in centimeters from the left. Next, all 16 scores of the VAS were into three categories: contentedness, calmness, and alertness following previous methods (*Bond and Lader, 1974*). HR, MAP, and VAS scores were corrected by calculating percentage change from the baseline measurement. Next, we applied 1-factor rmANOVAs (factor drug), on these scores for each of the remaining timepoints. Post hoc, pairwise t-tests were performed to test the effects of ATX and DNP vs. PLC.

We assessed the effects of drug on pupil diameter right before the onset of the behavioral tasks (at t=4 hr). We defined minimal (bright monitor background) and maximal (dark monitor background) pupil diameter as the average pupil size within the final 5 s of each presentation window of 15 s. Next, we performed similar rmANOVAs and post hoc t-tests on these raw (non-normalized) pupil diameters. To test the proportion of forced guesses of drug intake at the end of each session, we used proportion z-tests. Specifically, we calculated proportions of guesses about the nature of drug intake (PLC or pharmaceutical) under PLC. Then we calculated the same proportions for ATX and DNP and tested them against the PLC proportion with a proportion z- test.

### Statistical analysis of behavioral data

To investigate behavioral performance on this task, we used perceptual sensitivity (d') and bias (criterion) derived from SDT (*Green and Swets, 1966*) in addition to RTs. To test the effect of cue validity and drugs on these measures, we performed 3×2 factorial rmANOVAs (factors: drug and cue validity). We did not include drug order as a between-subject variable in our statistical models, because drug order was counter-balanced between participants. Bayesian rmANOVAs (uniform prior, default setting) were used in the case of null findings, to test support for the null-hypothesis (*JASP TEAM, 2022*). We expressed all effect sizes as $\eta_p^2$ . For t-tests we calculated these as follows:

$$\eta_p^2 = \frac{t^2}{t^2 + df} \qquad (4)$$

where t is the t-statistic and df the degrees of freedom. Effect sizes are considered small when $\eta_p^2$ = 0.01, medium when $\eta_p^2$ = 0.06, and large when $\eta_p^2$ = 0.14 (*Cohen, 1988*).

## Drift diffusion modeling

We constructed a DDM to gain insight into which parameters of the decision process were modulated by attention and drug condition (*Ratcliff and McKoon, 2008*). Specifically, we used the Python package HDDM to fit hierarchical Bayesian accuracy-coded regression models to RT distributions of correct and incorrect trials for every participant (*Wiecki et al., 2013*). In a first regression model, we allowed drift rate, non-decision time, and decision bound separation to vary with drug, cue validity, and their interaction. To estimate the effects of drug, we fitted these regression models separately for ATX (vs. PLC) and DNP (vs. PLC). We applied a weighted effect coding scheme, meaning that we coded our regressors as –1 and 1 (instead of dummy coding: 0 and 1). We weighted the regressors for cue validity according to the proportion of valid and invalid trials (–0.8 for invalid, 0.2 for valid), because valid and invalid trial counts were unbalanced. The effects of this model can be interpreted similarly as effects derived from an ANOVA. The Bayesian implementation of this model constrained single subject parameter estimates based on population estimates, making the model more robust to noise. Note that we also fitted unweighted effect coded versions of these models to verify the output of our weighted models, as well as two additional (weighted effect-coded) models to test whether drift rate variability was also modulated by cue validity and drug condition (*Figure 2—figure supplements 4 and 5*; *Murphy et al., 2014*).

## EEG-ERP analysis

All analyses were performed on the CSD-transformed EEG data. To assert how spatial attention and elevated baseline levels of neuromodulators shape visual perception, we looked at neural activity related to the processing of the visual input (cue and stimulus) as well as activity related to the sensory information accumulation process preceding the response. For statistical analyses, epoched data were downsampled to 128 Hz. We used distinct spatial ROIs for each of these events, based on previous literature. For cue processing, we used symmetrical electrode pairs O1/O2, PO3/PO4, and PO7/PO8 (*Capotosto et al., 2009*; *Kelly et al., 2006*; *Thut et al., 2006*), for processing of the visual stimulus, we used the symmetrical electrode pairs P7/P8 and P9/P10 (*Loughnane et al., 2016*; *Newman et al., 2017*; *Papaioannou and Luck, 2020*; *van Kempen et al., 2019*), and for response-locked evidence accumulation signals (i.e. CPP) we used electrodes CPz, Cp1, and Cp2 (*O'Connell et al., 2012*; *van Kempen et al., 2019*; *van Vugt et al., 2019*). To extract ERPs, we first normalized epochs by subtracting the average baseline activity –80 ms to 0 ms before cue onset (for cue-locked ERP) and stimulus onset (for stimulus-locked and response-locked ERP).

We calculated the peak amplitude of the CPP as the amplitude at the time of the response and the slope of the CPP by fitting a linear regression in a time-window ranging from –250 ms to 0 ms before the response. Note that the time-window used for CPP slope estimation was chosen on the basis of visual inspection of the grand average CPP and roughly coincided with previous time-windows used for slope estimation (*van Kempen et al., 2019*; *van Vugt et al., 2019*). Next, CPP peak amplitude and slope were tested for incorrect vs. correct and fast vs. slow response trials (median split) with two-sided pairwise t-tests and for high vs. low drift rate participants with independent t-tests (*Figure 3A–C*). Moreover, 2×3 (attention × drug) rmANOVAs and post hoc two-sided t-tests (drug vs. PLC) were used to test the effects of cue validity and drug on CPP slope and peak amplitude. We further used cluster-corrected rmANOVAs over time to establish the effects of target vs. non-target information, spatial attention, cue validity, and drug on cue-locked and stimulus-locked ERP data. Last, we extracted activity from significant clusters, to test for other main and interaction effects.

We analyzed stimulus-locked ERPs with a cluster-corrected permutation (10,000 permutations) 2 × 2 × 3 (cue × stimulus identity × drug) rmANOVA. For the cue-locked ERP (*Figure 5A*), we performed a cluster-corrected permutation (10,000 permutations) 3×2 (drug × hemisphere) rmANOVA.

## EEG-TF analysis

To test the effects of cue validity and drug condition on oscillatory brain dynamics, we first calculated the lateralization of TF power (contralateral-ipsilateral to cue) across all drug conditions and then

tested this lateralization against zero with cluster-corrected two-sided permutation tests (10,000 permutations; *Figure 5C*). Then, we extracted power from the alpha-band (8–12 Hz) and tested the effects of hemisphere (contralateral vs. ipsilateral to cue) and drug condition on power within this frequency band with a cluster-corrected permutation (10,000 permutations) 3×2 (drug × hemisphere) rmANOVA. Last, we extracted lateralized single trial alpha power from the late cluster plotted in *Figure 5D* and binned data according to prestimulus alpha power in two evenly sized bins. Then, for every bin and cue validity condition we calculated d' and tested the effects of drug, prestimulus alpha power bin, and cue validity on d' with a 3 × 2 × 2 rmANOVA (*Figure 5—figure supplement 1B*).

## Additional information

### Funding

| Funder | Grant reference number | Author |
| --- | --- | --- |
| HORIZON EUROPE European Research Council | ERC STG 715605 | Simon van Gaal |

The funders had no role in study design, data collection and interpretation, or the decision to submit the work for publication.

### Author contributions

Stijn A Nuiten, Conceptualization, Formal analysis, Investigation, Visualization, Methodology, Writing – original draft; Jan Willem de Gee, Formal analysis, Methodology, Writing – review and editing; Jasper B Zantvoord, Data curation, Project administration; Johannes J Fahrenfort, Conceptualization, Supervision, Methodology, Writing – review and editing; Simon van Gaal, Conceptualization, Supervision, Funding acquisition, Writing – original draft, Project administration

### Author ORCIDs

Stijn A Nuiten ⓘ https://orcid.org/0000-0002-9248-166X
Jan Willem de Gee ⓘ https://orcid.org/0000-0002-5875-8282
Johannes J Fahrenfort ⓘ https://orcid.org/0000-0002-9025-3436
Simon van Gaal ⓘ https://orcid.org/0000-0001-6628-4534

### Ethics

Human subjects: Informed consent and consent to publish was obtained from all participants. All procedures were approved by the Medical Ethical Committee of the Amsterdam Medical Centre (METC AMC), under project number NL64341.018.18.

Joint Public Review: https://doi.org/10.7554/eLife.87022.3.sa1
Author Response https://doi.org/10.7554/eLife.87022.3.sa2

## Additional files

### Supplementary files
• MDAR checklist

### Data availability

We have added source data for all figures and figure supplements. All raw data and analyses scripts (including pre-processing and statistical analyses) are available on FigShare, FigShare project (https://uvaauas.figshare.com/projects/Catecholaminergic_neuromodulation_and_selective_attention_jointly_shape_perceptual_decision-making/160049) with the following datasets https://doi.org/10.21942/uva.24624318.v1, https://doi.org/10.21942/uva.24624321.v1, https://doi.org/10.21942/uva.24624342.v1 and https://doi.org/10.21942/uva.24631050.v1.

The following datasets were generated:

| Author(s) | Year | Dataset title | Dataset URL | Database and Identifier |
|---|---|---|---|---|
| Nuiten SA | 2023 | Physiological and subjective experience data | https://doi.org/10.21942/uva.24624318.v1 | FigShare, 10.21942/uva.24624318.v1 |
| Nuiten SA | 2023 | Pupil data | https://doi.org/10.21942/uva.24624321.v1 | FigShare, 10.21942/uva.24624321.v1 |
| Nuiten SA | 2023 | Raw EEG data | https://doi.org/10.21942/uva.24624342.v1 | FigShare, 10.21942/uva.24624342.v1 |
| Nuiten SA | 2023 | Behavioral data | https://doi.org/10.21942/uva.24631050.v1 | FigShare, 10.21942/uva.24631050.v1 |

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
