## [Editor Report · eLife assessment]

This **important** study shows that pharmacologically enhanced catecholamine levels and increased voluntary spatial attention have overlapping as well as dissociable effects on performance on a visuospatial attention task and corresponding EEG markers. The findings provide **solid** evidence regarding how neuromodulatory arousal and selective spatial attention jointly shape perceptional decision-making.

---

## [Referee Report · Joint Public Review]

The authors aimed to contrast the effects of pharmacologically enhanced catecholamine and acetylcholine levels versus the effects of voluntary spatial attention on decision making in a standard spatial cuing paradigm. Meticulously reported, the authors show that atomoxetine, a norepinephrine reuptake inhibitor, and cue validity both enhance model-based evidence accumulation rate, but have several distinct effects on EEG signatures of pre-stimulus cortical excitability, evoked sensory EEG potentials and perceptual evidence accumulation. The results are based on a reasonable sample size (N=28) and state-of-the art modeling and EEG methods.

The authors' EEG findings provide solid evidence for the overall conclusion that selective attention and neuromodulatory systems shape perception in "similar, unique, and interactive" ways. This is an important conclusion because neuromodulatory systems and selective spatial attention are both known to regulate the neural gain of task-relevant single neurons and neural networks. Apparently, these effects on neural gain affect decision making in partly overlapping and partly dissociable ways.

The effects of donepezil, a cholinesterase inhibitor, were generally less strong than those of atomoxetine, and in various analyses went in the opposite direction. The authors fairly conclude that more work is necessary to determine the effects of cholinergic neuromodulation on perceptual decision making.

---

## [Author Response]

The following is the authors’ response to the original reviews.

We were pleased with seeing our work published as a Reviewed Preprint online so swiftly. Now, we would like to take the opportunity to include our responses to the comments made by the reviewers into the Reviewed Preprint and also submit a revised version of the manuscript, in which we have incorporated and addressed the reviewers’ comments.

We believe that our revisions have significantly improved the quality of the manuscript. Specifically, we have described our results more precisely and explained certain decisions that were made in the analysis pipeline more clearly. For example, Figure 4 was improved substantially, by incorporating a schematic representation of how ERP traces were extracted from neural data. Furthermore, we have added three paragraphs in the Discussion where we elaborate on (1) the two observed interaction effects between attention and drug condition, (2) the relation between behavioral, computational, and neural effects, and

1. the statistical robustness of our findings. As such, we believe our interpretation of the results and their robustness now more faithfully represents our observations.

Moreover, we have incorporated the Supplementary Information and Figures, initially presented as a separate section of the manuscript, to the main manuscript and its accompanying supplementary figures. Thereby, the structure of the paper now better follows the eLife format. As a result, some of the previously included supplementary figures are now described in text of the main manuscript.

**Reviewer #1 comments:**
In the results section on page 6, the authors conclude that "Attention and ATX both enhanced the rate of evidence accumulation towards a decision threshold, whereas cholinergic effects were negligible." I believe "negligible" is wrong here: the corresponding effects of donepezil had p-values of .09 (effect of donepezil on drift rate), .07 (effect of donepezil on the cue validity effect on drift rate) and .09 (effect of donepezil on non-decision time), and were all in the same direction as the effects of atomoxetine, and would presumably have been significant with a somewhat larger sample size. I would say the effects of donepezil were "in the same direction but less robust" (or at the very least "less robust") instead of "negligible".

We agree with the reviewer that ‘negligible’ may not properly capture the effects of DNP on DDM parameter estimates. Although we do feel that caution is warranted in interpreting the effects of DNP on computational parameter estimates, we have now described these effects in line with the reviewer’s suggestion: in the same direction as the effects of ATX, but not (or less) statistically robust.

"In the results section on page 8, the authors conclude that "Summarizing, we show that drug condition and cue validity both affect the CPP, but they do so by affecting different features of this component (i.e. peak amplitude and slope, respectively)." This conclusion is a bit problematic for two reasons. First, drug condition had a significant effect not only on peak amplitude but also on slope. Second, cue validity had a significant effect not only on slope but also on peak amplitude. It may well be that some effects were more significant than others, but I think this does not warrant the authors' conclusion.

Indeed, we observed that cue validity affected both CPP peak amplitude and slope and some effects were more significant than others. As such, we agree with the reviewer that the conclusion that cue validity and drug condition affect different features of the CPP was too strongly formulated. We have changed this statement in the manuscript to reflect the observed data pattern more appropriately. We would however like to point out that this does not undermine our main conclusion. Spatial attention and drug condition showed only limited interaction effects in terms of behavior and neural data and their effects on occipital activity were separable in terms of timing and spatial profile. Therefore, our conclusion that catecholamines and spatial attention jointly shape perceptual decision-making remains valid.

In the discussion section on page 11, the authors conclude that "First, although both attention and catecholaminergic enhancement affected centro-parietal decision signals in the EEG related to evidence accumulation (O'Connell et al., 2012; Twomey et al., 2015), attention mainly affected the build-up rate (slope) whereas ATX increased the amplitude of the CPP component (Figure 3D-F)." As I wrote above, I believe it is not correct that "attention mainly affected the build-up rate or slope", given that the effect of cue-validity on CPP slope was also significant. Also, while the authors' data do support the conclusion that ATX increased the amplitude and not the slope of the CPP component, a previous study in humans found the opposite: ATX increased the slope but did not affect the peak amplitude of the CPP (Loughnane et al 2019, JoCN, https://pubmed.ncbi.nlm.nih.gov/30883291). Although the authors cite this study (as from 2018 instead of 2019), they do not draw attention to this important discrepancy between the two studies. I encourage the authors to dedicate some discussion to these conflicting findings.

We thank the reviewer for spotting this error, we cited the preprint version (from 2018) of Loughnane and colleagues and not the published JoCN paper (from 2019). We have changed this in the updated version of the manuscript. We further thank the reviewer for asking about this interesting discrepancy between our observation that ATX increased CPP peak amplitude in absence of slope effects and the observation by Loughnane et al. (2019, JoCN) that ATX increased CPP slope, but not amplitude. We first would like to point out that the peak amplitude effect in Loughnane et al. (2019) was in the same direction as our reported effect, with numerically higher peak amplitudes for ATX compared to PLC (Figure 2A – right panel in Loughnane et al., 2019). However, as their omnibus main effect of drug condition on CPP peak amplitude was not significant, they did not provide statistics for a pairwise comparison of ATX and PLC in terms of CPP peak amplitude, which makes it hard to compare the effects directly. Regardless, Loughnane et al. (2019) did observe an effect on CPP slope, whereas we did not. Speculatively, this difference could be related to the behavioral tasks that were used in both studies. Below we have added a new paragraph from the Discussion in which we elaborate on this more.

In Discussion, page 15:

Here, we demonstrated that response accuracy and response speed are differentially represented in the CPP, with correct vs. erroneous responses resulting in a higher slope and peak amplitude, whereas fast vs. slow responses are only associated with increased slopes (Figure 3A-B). Speculatively, the specific effect of any (pharmacological) manipulation on the CPP may depend on task-setting. For example, Loughnane et al. (2019) used a visual task on which participants did not make many errors (hit rate>98%, no false alarms), whereas we applied a task in which participants regularly made errors (roughly 25% of all trials). Possibly, the effects of ATX from Loughnane et al. (2019) in terms of behavior (RT effect, not accuracy/d’) and CPP feature (slope effect, not peak) may therefore have been different from the effects of ATX we observed on behavior (d’ effect, not RT) and CPP feature (peak effect, not slope). Regardless, when we compared subjects with high and low drift rates (Figure 3C), we observed that both CPP slope and CPP peak were increased for the high vs. low drift group (independent of the drug or attentional manipulation). This indicates that both CPP slope and CPP peak were associated with drift rate from the DDM. Clearly, more work is needed to fully understand how evidence accumulation unfolds in neural systems, which could consequently inform future behavioral models of evidence accumulation as well.

On page 12 and page 14 the authors suggest a selective effect of ATX on *tonic* catecholamine activity, but to my knowledge the exact effects of ATX on phasic vs. tonic catecholamine activity are unknown. Although microdialysis studies have shown that a single dose of atomoxetine increases catecholamine concentrations in rodents, it is unknown whether this reflects an increase in tonic and/or phasic activity, due to the limited temporal resolution of microanalysis. Thus, atomoxetine may affect tonic and/or phasic catecholamine activity, and which of these two effects dominates is still unknown, I think.

We agree with the reviewer that the direct effects of ATX on tonic versus phasic catecholaminergic activity are not clear as initially stated in the manuscript. Equally problematic, previous work has demonstrated that changes in tonic neuromodulation shape evoked neuromodulatory discharge (Aston-Jones & Cohen, 2005, Annu. Rev. Neurosci; Knapen et al., 2016, PLoS ONE). As such, any effect of ATX on tonic neuromodulatory drive would probably have affected phasic catecholaminergic responses as well, although this claim will have to be experimentally addressed. We think that because of the close relation between tonic and phasic neuromodulation, it may indeed be better to refrain from the simplistic interpretation that ATX (and DNP) solely and specifically affects tonic neuromodulation. We have used more neutral language in that regard in the updated version of the manuscript, for example by only mentioning elevated neuromodulator levels (not specifying tonic or phasic). Moreover, we have extended a part of our previous Discussion, to elaborate this issue in more detail. An excerpt of this paragraph, consisting of previous and newly added text, can be seen below.

In Discussion, page 14:

In contrast with recent work associating catecholaminergic and cholinergic activity with attention by virtue of modulating prestimulus alpha-power shifts (Bauer et al., 2012; Dahl et al., 2020, 2022) and attentional cue-locked gamma-power (Bauer et al., 2012; Howe et al., 2017), the current work shows that the effects of neuromodulator activity are relatively global and non-specific, whereas the effects of spatial attention are more specific to certain locations in space. Our findings are, however, not necessarily at odds with these previous studies. Most recent work associates phasic (event-related) arousal with selective attention (for reviews see: Dahl et al., 2022; Thiele & Bellgrove, 2018). For example, cue detection in visual tasks is known to be related to cholinergic transients occurring after cue onset (Howe et al., 2017; Parikh et al., 2007). Contrarily, in our work we aimed to investigate the effects of increased baseline levels of neuromodulation by suppressing the reuptake of catecholamines and the breakdown of acetylcholine throughout cortex and subcortical structures. Tonic and phasic neuromodulation have previously been shown to differentially modulate behavior and neural activity (de Gee et al., 2014, 2020, 2021; McGinley et al., 2015; McGinley, Vinck, et al., 2015; van Kempen et al., 2019). Note, however, that it is difficult to investigate causal effects of tonic neuromodulation in isolation of changes in phasic neuromodulation, mostly because phasic and tonic activity are thought to be anti-correlated, with lower phasic responses following high baseline activity and vice versa (Aston- Jones & Cohen, 2005; de Gee et al., 2020; Knapen et al., 2016). As such, pharmacologically elevating tonic neuromodulator levels may have resulted in changes in phasic neuromodulatory responses as well. Concurrent and systematic modulations of tonic (e.g. with pharmacology) and phasic (e.g. with accessory stimuli; Bruel et al., 2022; Tona et al., 2016) neuromodulator activity may be necessary to disentangle the respective and interactive effects of tonic and phasic neuromodulator activity on human perceptual decision-making.

**Reviewer #2 comments:**
The main weakness of the paper lies in the strength of evidence provided, and how the results tally with each other. To begin with, there are a lot of significance tests performed here, increasing the chances of false positives. Multiple comparison testing is only performed across time in the EEG results, and not across post-hoc comparisons throughout the paper. In and of itself, it does not invalidate any result per se, but it does colour the interpretation of any results of weak significance, of which there are quite a few. For example, the effect of Drug on d' and subsequent post-hoc comparisons, also effect of ATX on CPP amplitude and others.

We agree with the reviewer that the statistical evidence for some of the results presented in this study is limited. This issue mostly concerns the effects of the pharmacological manipulation (effects of attention were strong and robust), which is unfortunately often the case given the high inter-individual variability in responses to pharmaceutical agents. We have added a paragraph to the Discussion in which we discuss this limitation of the current study. Furthermore, we discuss our findings in the context of previous work, thereby showing that - although not always robust- most of the reported drug effects were in the direction that could be expected based on previous literature. We have pasted that paragraph below.

In Discussion, pages 16:

Although the effects of the attentional manipulation were generally strong and robust, the statistical reliability of the effects of the pharmacological manipulation was more modest for some comparisons. This may partly be explained by high inter-individual variability in responses to pharmaceutical agents. For example, initial levels of catecholamines may modulate the effect of catecholaminergic stimulants on task performance, as task performance is supposed to be optimal at intermediate levels of catecholaminergic neuromodulation (Cools & D’Esposito, 2011). While acknowledging this, we would like to highlight that many of the observed effects of ATX were in the expected direction and in line with previous work. First, pharmacologically enhancing catecholaminergic levels have previously been shown to increase perceptual sensitivity (d’) (Gelbard-Sagiv et al., 2018), a finding that we have replicated here. Second, methylphenidate (MPH), a pharmaceutical agent that elevates catecholaminergic levels as well, has been shown to increase drift rate as derived from drift diffusion modeling on visual tasks (Beste et al., 2018) in line with our ATX observations. Third, a previous study using ATX to elevate catecholaminergic levels observed that ATX increased CPP slope (Loughnane et al., 2019). Although in our case ATX increased the CPP peak and not its slope, this provide causal evidence that centro-parietal ERP signals related to sensory evidence accumulation are modulated by the catecholaminergic system (Nieuwenhuis et al., 2005). Fourth, we observed that elevated levels of catecholamines affected stimulus driven occipital activity relatively late in time and close to the behavioral response, which resonates with previous observations (Gelbard-Sagiv et al., 2018). Finally, ATX had robust effects on physiological responses (heart rate, blood pressure, pupil size), cue-locked ERP signals and oscillatory power dynamics in the alpha-band, leading up to stimulus presentation. We concur, however, that more work is needed to firmly establish how (various forms of) attention and catecholaminergic neuromodulation affect perceptual decision-making.

The lack of an overall RT effect of Drug leaves any DDM result a little underwhelming. How do these results tally? One potential avenue for lack of RT effect in ATX condition is increased drift rate but also increased non-decision time, working against each other. However, it may be difficult to validate these results theoretically.

As the reviewer remarks, an increase in performance/d’ in absence of any RT effects can be algorithmically explained by a combination of increased drift rate and prolonged non-decision time. This is indeed what we observed for ATX. Non-decision time is generally thought to reflect the time necessary for stimulus encoding and motor execution and as such is seen as separate from the evidence-accumulation decision process. We deem it possible that ATX simultaneously prolonged stimulus encoding/motor execution (reflected in changes in non-decision time) and fastened evidence accumulation (reflected in changes in drift rate). Although our neural data did not provide evidence for this claim, previous work has demonstrated that increased baseline (pupil-linked) arousal/neuromodulation is associated with a decreased build-up rate of a neural signal associated with motor execution (β-power over motor cortex, Van Kempen et al., 2019, eLife), potentially linking increased non-decision time under ATX to slowing down of motor execution processes. The same authors also report relationships between baseline (pupil-linked) arousal/neuromodulation and activity over occipital and centroparietal cortices, respectively associated with sensory processing and sensory evidence accumulation, suggesting that baseline neuromodulation may affect all stages leading up to a decision (sensory processing, evidence accumulation and motor execution). Note also that the attentional manipulation seems to simultaneously increase drift rate and shorten non-decision time in our case, as one would expect (Figure 2E, Figure 2 – Supplements 4&5).

There is an interaction between ATX and Cue in terms of drift rate, this goes against the main thesis of the paper of distinct and non-interacting contributions of neuromodulators and attention. This finding is then ignored. There is also a greater EDAN later for ATX compared to PLA later in the results, which would also indicate interaction of neuromodulators and attention but this is also somewhat ignored.

There are indeed some interesting interaction effects between ATX and spatial attention (cue), as pointed out by the reviewer. However, we did also observe striking differences in the effects of ATX and attention on stimulus-locked occipital activity (in timing and spatial specificity) as well as independent (main) effects on CPP amplitude and pre-stimulus alpha power. Therefore, throughout the paper we tried to carefully describe the effects of attention and ATX as largely independently and jointly modulating perceptual decision-making, while at the same time highlighting the interaction effects that we observed, where present. We have highlighted the effects the reviewer refers to even more explicitly in a separate paragraph that we added to the discussion, pasted below.

In Discussion, page 13-14:

We did observe two striking interaction effects between the catecholaminergic system and spatial attention. First, effects of attention on drift rate were increased under catecholaminergic enhancement (Figure 2D). Although this interaction effect was not reflected in CPP slope/peak amplitude, this does suggest that catecholamines and spatial attention might together shape sensory evidence accumulation in a non-linear manner. Second, the amplitude of the cue-locked early lateralized ERP component (resembling the EDAN) was increased under ATX as compared to PLC. The underlying neural processes driving the EDAN ERP, as well as its associated functions, have been a topic of debate. Some have argued that the EDAN reflects early attentional orienting (Praamstra & Kourtis, 2010) but others have claimed it is mere a visually evoked response and reflects visual processing of the cue (Velzen & Eimer, 2003). Thus, whether this effect reflects a modulation of ATX on early attentional processes or rather a modulation of early visual responses to sensory input in general is a matter for future experimentation.

The CPP results are somewhat unclear. Although there is an effect of ATX on drift rate algorithmically, there is no effect of ATX on CPP slope. On the other hand, even though there is no effect of DNP on drift rate, there is an effect of DNP on CPP slope. Perhaps one may say that the effect of DNP on drift rate trended towards significance, but overall the combination of effects here is a little unconvincing. In addition, there is an effect of ATX on CPP amplitude, but how does this tally with behaviour? Would you expect greater CPP amplitude to lead to faster or slower RTs? The authors do recognise this discrepancy in the Discussion, but discount it by saying the relationship between algorithmic and CPP parameters in terms of DDM is unclear, which undermines the reasoning behind the CPP analyses (and especially the one correlating CPP slope with DDM drift rate).

We thank the reviewer for pointing out this dissociation of drug effects in terms of the algorithmic (DDM) and neural (CPP) ‘implementations’ of the evidence accumulating process underlying perceptual decisions. We have added a new paragraph to the discussion where we interpret the effects of ATX on the neural and algorithmic levels of evidence accumulation. Below we have pasted that paragraph:

In Discussion, page 14-15:

We reported attentional and neuromodulatory effects on algorithmic (DDM, Figure 2) and neural (CPP, Figure 3) markers of sensory evidence accumulation. Recent work has started to investigate the association of these two descriptors of the accumulation process, aiming to uncover whether neural activity over centroparietal regions reflects evidence accumulation, as proposed by computational accumulation-to-threshold models (Kelly & O’Connell, 2015; O’Connell et al., 2018; O’Connell & Kelly, 2021; Twomey et al., 2015). Currently, the CPP is often thought to reflect the decision variable, i.e. the (unsigned) evidence for a decision (Twomey et al., 2015), and consequently its slope should correspond with drift rate, whereas its amplitude at any time should correspond with the so-far accumulated evidence. As -computationally- the decision is reached when evidence crosses a decision bound (the threshold), it may be argued that the peak amplitude of the CPP (roughly) corresponds with the decision boundary. This seems to contradict our observation that (1) ATX modulated drift rate, but not CPP slope and (2) ATX did not modulate boundary separation, but did modulate CPP peak. Note, however, that previous studies using pharmacology or pupil-linked indexes of (catecholaminergic) neuromodulation have also demonstrated effects on both CPP peak (van Kempen et al., 2019) and CPP slope (Loughnane et al., 2019).

The posterior component effects are problematic. The main issue is the lack of clarification of and justification for the choice of posterior component. The analysis is introduced in the context of the target selection signal the N2pc/N2c, but the component which follows is defined relative to Cue, albeit post-target. Thus this analysis tells us the effect of Cue on early posterior (possibly) visual ERP components, but it is not related to target selection as it is pooled across target/distractor. Even if we ignore this, the results themselves wrt Drug lack context. There is a trending lower amplitude for ATX at later latencies at temporo-parietal electrodes, and more positive for DNP, relative to PLA. Is this what one would expect given behaviour? This is where the issue of correct component identification becomes critical in order to inform any priors on expected ERP results given behaviour.

We thank the reviewer for raising this issue with the occipital ERP analysis, allowing us to clarify our decisions regarding the analyses and our interpretations of the results. First, the selection of electrodes was based on, and identical to, previous studies investigating lateralized target selection signals in visual tasks containing bilateral visual stimuli (Loughnane et al., 2016; Newman et al., 2017; Papaioannou & Luck, 2020; van Kempen et al., 2019). Second, the ERPs were defined relative to both the direction of the cue as well as the location of the target. As cue direction and target location were not always congruent (cue validity=80%), we could adopt a 2x2 (cue direction x stimulus identity) design for our ERP analyses (we are ignoring drug condition for explanation purposes). For example, for validly cued target trials we extracted two ERP traces: (1) from the hemisphere contralateral to both the cue and the target stimulus (representing processing of cued target stimulus) and (2) from the hemisphere ipsilateral to the cue and the target stimulus (representing processing of non-cued noise stimulus). However, for invalidly cued trials, ERP traces were extracted from (3) the hemisphere contralateral to cue direction and ipsilateral to the target stimulus (reflecting processing of cued noise stimuli) as well as (4) from the hemisphere ipsilateral to cue direction but contralateral to the target stimulus (reflecting processing of non-cued target stimuli). By defining our ERPs as such, we were able to gauge effects of cue direction (reflecting general shifts in attention), stimulus identity (reflecting target vs. noise selection processes) and their interaction (reflecting cue validity) on activity over occipito-temporal activity. Third, we did not pool data (across target/noise stimuli) for statistical analyses, but only for visualization purposes. To clarify how we extracted ERP traces, we have changed Figure 4 substantially. The updated figure now contains a schematic of how these four distinct ERP traces (cue x stimulus identity) were extracted from neural activity. Moreover, for clarity sake, we now show all 12 ERP traces (3x2x2, drug condition x cue direction x stimulus identity) as well as the three main effects that we observed after performing a 3x2x2 repeated measures (rm)ANOVA over time.

We observed robust (cluster-corrected) effects of cue direction (not validity) on early occipital activity (Fig. 4C – left panel) and of stimulus identity (target/noise) and drug condition on later occipital activity (Fig. 4C – middle and right panel). These results crucially highlight the different temporal (early/late) and spatial (lateralized/not lateralized) profiles of cue, target and drug effects on occipital activity. Moreover, we observed a specific order of drug effects on late occipital activity (DNP>PLC>ATX). The behavioral relevance of this pattern of effects remains elusive. Although the effects of drug condition coincide in time with those of target selection (i.e. when activity contralateral and ipsilateral to the target stimulus was different), the effects of drug were bilateral, meaning that occipito-temporal activity related to the processing of the target (task-relevant) stimulus and non-target (task-irrelevant) stimulus was equally modulated by these pharmaceutical agents. One might argue that these effects show that neither ATX nor DNP modulated the signal-to-noise ratio (SNR), a feature that describes how well relevant stimulus information (signal) can be discerned from irrelevant information (noise). Although it may be tempting to extrapolate this finding to behavior, by suggesting that on the basis of these drug effect neither ATX nor DNP could have modulated d’ (behavioral measure describing how well signal is separated from noise), we would like to point out that our behavioral task specifically concerned a discrimination task about the (orientation of the) target stimulus in which the difference between signal and noise was only relevant for localization purposes and thus has a less direct relation with task performance. As such it is difficult to grasp how the modulation of late occipito-temporal activity by ATX and DNP relates to their behavioral effects. Moreover, the bilateral effect of both ATX and DNP also suggests an absence of interaction effects between drug conditions and visuo-spatial attention, as the effects of ATX/DNP were similar across all cue and target identity conditions.